# Remote Sensing of Cloud Droplet Radius Profiles using solar reflectance from cloud sides. Part I: Retrieval development and characterization

Florian Ewald[1,2], Tobias Zinner[1], Tobias Kölling[1], and Bernhard Mayer[1]

[1]Ludwig Maximilians Universität, Meteorologisches Institut, München, Germany
[2]Deutsches Zentrum für Luft und Raumfahrt, Institut für Physik der Atmosphäre, Oberpfaffenhofen, Germany

**Correspondence:** Florian Ewald (florian.ewald@dlr.de)

**Abstract.** Convective clouds play an essential role for Earth's climate as well as for regional weather events since they have a large influence on the radiation budget and the water cycle. In particular, cloud albedo and the formation of precipitation are influenced by aerosol particles within clouds. In order to improve the understanding of processes from aerosol activation, over cloud droplet growth to changes in cloud radiative properties, remote sensing techniques become more and more important. While passive retrievals for spaceborne observations have become sophisticated and commonplace to infer cloud optical thickness and droplet size from cloud tops, profiles of droplet size have remained largely uncharted territory for passive remote sensing. In principle they could be derived from observations of cloud sides, but faced with the small-scale heterogeneity of cloud sides, 'classical' passive remote sensing techniques are rendered inappropriate. In this work the feasibility is demonstrated to gain new insights into the vertical evolution of cloud droplet effective radius by using reflected solar radiation from cloud sides. Central aspect of this work on its path to a working cloud side retrieval is the analysis of the impact unknown cloud surface geometry has on effective radius retrievals. This study examines the sensitivity of reflected solar radiation to cloud droplet size, using extensive 3D radiative transfer calculations on the basis of realistic droplet size resolving cloud simulations. Furthermore, it explores a further technique to resolve ambiguities caused by illumination and cloud geometry by considering the surrounding of each pixel. Based on these findings, a statistical approach is used to provide an effective radius retrieval. This statistical effective radius retrieval is focused on the liquid part of convective water clouds, e.g., Cumulus mediocris, Cumulus congestus and Trade-wind cumulus, which exhibit well-developed cloud sides. Finally, the developed retrieval is tested using known and unknown cloud side scenes to analyze its performance.

## 1 Current state of passive remote sensing of clouds

There exist various methods to infer optical properties (e.g., optical thickness and cloud droplet effective radius) from observation of cloud tops using information about the scattered and absorbed radiation in the solar spectrum (e.g., Plass and Kattawar, 1968; King, 1987). Phase detection is the first step for every cloud property retrieval. Spectral absorption differences in the near-infrared or brightness temperature differences in the thermal infrared are commonly used to distinguish between liquid

water and ice (e.g., Nakajima and King, 1990). Various operational techniques exist to retrieve microphysical cloud properties like cloud thermodynamic phase and effective particle size (e.g., Han et al., 1994; Platnick et al., 2001; Roebeling et al., 2006).

Remote sensing of cloud and aerosol parameters is mostly done by use of multi-spectral sensors, i.e., using only a limited number of spectral bands. Common examples of spaceborne imagers are the Advanced Very High Resolution Radiometer (AVHRR), the Moderate Resolution Imaging Spectroradiometer (MODIS), and the Spinning Enhanced Visible Infrared Imager (SEVIRI). However, there are concerns about measurement artifacts influencing retrievals of aerosol and cloud properties caused by small-scale cloud inhomogeneity which are unresolved by the coarse spatial resolution of spaceborne platforms (Zinner and Mayer, 2006; Marshak et al., 2006b; Varnai and Marshak, 2007).

Non-imaging systems like the Solar Spectral Flux Radiometer (SSFR, Pilewski et al., 2003) or the Spectral Modular Airborne Radiation measurement sysTem (SMART, Wendisch et al., 2001; Wendisch and Mayer, 2003) were used for cloud remote sensing from ground (McBride et al., 2011; Jäkel et al., 2013) or aircraft (Ehrlich et al., 2008; Eichler et al., 2009; Schmidt et al., 2007).

**Scientific objectives and scope of this work**

In order to observe the vertical development of convective clouds' microphysics, Marshak et al. (2006a) and Martins et al. (2011) proposed cloud side scanning measurements while Zinner et al. (2008) and Ewald et al. (2013) presented concrete steps towards a cloud side retrieval for profiles of phase and particle size. Similar to previous satellite retrievals they propose to use solar radiation in the near-visible to near-infrared spectral regions reflected by cloud sides. Especially the vertical dimension of these observations should reflect aspects of cloud-aerosol-interaction as well as the mixing of cloudy and ambient air (Martins et al., 2011; Rosenfeld et al., 2012). However, the retrieval of cloud microphysical profiles demands a high spatial resolution on the order of 100 m or better. In turn, the high spatial resolution necessitates a method to consider 3D radiative transfer effects.

Albeit sophisticated, the studies of Zinner et al. (2008) and Ewald et al. (2013) are limited to an idealized geometry and simplified cloud microphysics. First, they focus on a space-like perspective for a fixed viewing zenith and scattering angle above the cloud field where sun and sensor have the same azimuth. Therefore, their studies lack the varying geometries of an airborne perspective and avoid the challenge to identify suitable observation positions within the cloud field. Moreover, the spatial resolution of their model cloud fields of $250\,\mathrm{m}$ is still rather coarse for an airborne perspective of cloud sides. Second, the effective radius is only parameterized in their studies. For all cloud fields, the effective radius profile is calculated by using a sub-adiabatic ascent of one air parcel in the context of a fixed cloud condensation nuclei (CCN) concentration. Finally, the approach was not tested for the potential bias to always detect larger effective radii with increasing cloud height; a potential pitfall that could be caused by the prior information contained in the forward calculations.

Since the diverse perspectives and the high spatial resolution of airborne cloud side measurements hampered the application of the approach presented by Zinner et al. (2008) and Ewald et al. (2013) until now, the present work will extend and test their ideas in the context of an airborne perspective. In the course of this part 1, the following scientific objectives will be addressed:

1. Extend the existing approach to realistic airborne perspectives and develop methods to test the sensitivity of reflected radiances from cloud sides to cloud droplet radius, where the observer position is located within the cloud field.

2. Investigate and mitigate 3D radiative effects which can interfere with the proposed cloud side remote sensing technique.

3. Test the approach in the context of realistic and explicit cloud microphysics with a specific focus on potential biases caused by the prior contained in the forward calculations.

The target of this work is the liquid part of convective water clouds, e.g., Cumulus mediocris, Cumulus congestus and Trade-wind cumulus, which exhibit well-developed cloud sides. During September 2014, images of such cloud sides were acquired with the **spec**trometer of the **M**unich **A**erosol **C**loud **S**canner (specMACS, Ewald et al., 2016) over the Amazonian rainforest near Manaus, Brazil. The measurements were performed during the ACRIDICON-CHUVA campaign (Wendisch et al., 2016) during which the specMACS instrument was deployed on the German research aircraft HALO (Krautstrunk and Giez, 2012), mounted in a side-looking configuration. The campaign focused on aerosol-cloud-precipitation interactions over the Amazon rain forest. More specific, the campaign investigated the impact of wildfire aerosols on cumulus clouds and on their later development into deep convection. During the campaign flights, the aerosol background and the small-scale convection in their early stages was probed in low-level flight legs between $1\,\mathrm{km}$ to $3\,\mathrm{km}$ altitude. At cloud base level, mean CCN concentrations ranged between $250\,\mathrm{cm}^{-3}$ and $2000\,\mathrm{cm}^{-3}$ (Andreae et al., 2018). The specMACS measurements were done of cumulus clouds in a distance of $2\,\mathrm{km}$ to $6\,\mathrm{km}$ and with top heights between $1.5\,\mathrm{km}$ and $3\,\mathrm{km}$. Subsequently, vertical profile flights were performed to measure the microphysical properties of the developing convection in-situ. This manuscript (Part 1) develops a statistical effective radius retrieval for these non-glaciated cumulus clouds which where measured during the low-level flights. Part 2 of this work presents the application to airborne specMACS data collected during the ACRIDICON-CHUVA and comparison to in-situ measurements.

This study is organized as follows: Section 2 shortly recapitulates established methods and introduces the new cloud model data set with explicit cloud microphysics. New methods to select suitable cloud sides and connect 3D radiances with 3D cloud microphysics will be described in Section 3. In Section 4, the sensitivity of reflected radiances to cloud droplet radii is examined for a simple, spherical cloud geometry, before moving the focus to the more realistic cloud side scenes. With the obtained insights, a method is developed to mitigate 3D radiative effects by using additional information from surrounding pixels. The extensive three-dimensional (3D) radiative transfer simulations of cloud sides, which form the basis of the statistical effective radius retrieval, is described in Section 5. In contrast to previous studies, different aerosol backgrounds are now also considered. For the retrieval, the results for different CCN concentrations are combined within one lookup table to be independent from a-priori knowledge of $N_{\mathrm{CCN}}$. Finally, the developed retrieval is tested in Section 6 with unknown scenes of cloud sides and different aerosol backgrounds. Furthermore, the retrieval is analyzed for potential biases.

## 2 Models

### 2.1 Statistical approach

The derivation of vertical profiles of cloud microphysics from radiance reflected by cloud sides is a strongly under-determined problem. The statistical approach tries to provide a probability for a specific cloud microphysical state (e.g., effective radius) where a deterministic inversion is impossible due to ambiguities caused by an unknown cloud geometry. This work will follow the approach proposed by Marshak et al. (2006a) and Zinner et al. (2008) who developed a statistical method to account for three-dimensional radiative effects on complex-shaped cloud sides. In their studies, a large number of 3D radiance simulations of cloud data sets provide a database for a statistical effective radius retrieval.

More specific, a forward model is used to perform an ensemble of radiative transfer calculations to estimate the joint probability $p_{\mathrm{fwd}}(L_{0.87}, L_{2.10}, r_{\mathrm{eff}})$ to observe the joint occurrence of radiances $L_{0.87}$, $L_{2.10}$ and effective radius $r_{\mathrm{eff}}$. The likelihood $p(L_{0.87}, L_{2.10}|r_{\mathrm{eff}})$ to observe radiances $L_{0.87}$ and $L_{2.10}$ for a specific effective radius $r_{\mathrm{eff}}$ is obtained when the joint probability is normalized with the number of calculations for $r_{\mathrm{eff}}$, the marginal probability $p_{\mathrm{fwd}}(r_{\mathrm{eff}})$. Subsequently, *Bayes' Theorem* is applied to obtain the posterior probability $p(r_{\mathrm{eff}}|L_{0.87}, L_{2.10})$ which solves the inverse problem to retrieve the most likely effective radius $r_{\mathrm{eff}}$ when radiances $L_{0.87}$ and $L_{2.10}$ are observed.

### 2.2 Monte Carlo approximation

When no analytical expression for the likelihood probability is available, Monte Carlo sampling from the joint distribution can be used to approximate the likelihood and posterior probability (Mosegaard and Tarantola, 1995). The sampling via the radiative transfer model yields a histogram $n(L_{0.87}, L_{2.10}, r_{\mathrm{eff}})$ of the frequency of observed radiances $L_{0.87}$ and $L_{2.10}$ and the corresponding effective radius $r_{\mathrm{eff}}$. With the histogram $n$ as a very simple non-parametric density estimator (Scott et al., 1977), the following relation between the histogram $n$ and the joined probability $p_{\mathrm{fwd}}(L_{0.87}, L_{2.10}, r_{\mathrm{eff}})$ and marginal probability $p_{\mathrm{fwd}}(r_{\mathrm{eff}})$ can be made:

$$p_{\mathrm{fwd}}(L_{0.87}, L_{2.10}, r_{\mathrm{eff}}) = \frac{1}{N} n(L_{0.87}, L_{2.10}, r_{\mathrm{eff}}) \tag{1}$$

$$p_{\mathrm{fwd}}(r_{\mathrm{eff}}) = \frac{1}{N} n(r_{\mathrm{eff}}). \tag{2}$$

Here, the number of radiative transfer results $N$ needs to be large enough for a successful estimation of these two probabilities. Simultaneously, the forward simulation has to cover all values expected in the real world application. With the likelihood probability $p(L_{0.87}, L_{2.10}|r_{\mathrm{eff}})$ as a conditional probability, it can be written as the quotient of the joined probability $p_{\mathrm{fwd}}(L_{0.87}, L_{2.10}, r_{\mathrm{eff}})$ and $p_{\mathrm{fwd}}(r_{\mathrm{eff}})$ from Equation (1) and Equation (2):

$$p(L_{0.87}, L_{2.10}|r_{\mathrm{eff}}) = \frac{p_{\mathrm{fwd}}(L_{0.87}, L_{2.10}, r_{\mathrm{eff}})}{p_{\mathrm{fwd}}(r_{\mathrm{eff}})} \tag{3}$$

$$p(r_{\mathrm{eff}}|L_{0.87}, L_{2.10}) = \frac{p(L_{0.87}, L_{2.10}|r_{\mathrm{eff}})\, p_{\mathrm{pr}}(r_{\mathrm{eff}})}{\int p(L_{0.87}, L_{2.10}|r_{\mathrm{eff}})\, p_{\mathrm{pr}}(r_{\mathrm{eff}})\, dr_{\mathrm{eff}}} \tag{4}$$

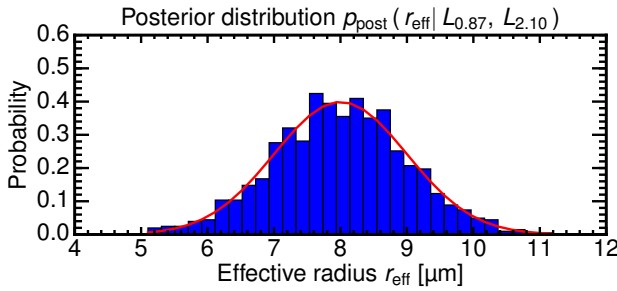

**Figure 1.** Approximation of a posterior pdf (red) by Monte Carlo Sampling (blue).

In Equation (3), the distribution of $r_{\text{eff}}$ in the radiative transfer ensemble is removed by the normalization with the marginal probability $p_{\text{fwd}}(r_{\text{eff}})$. In a final step, the likelihood probability can be used with an arbitrary prior $p_{\text{pr}}(r_{\text{eff}})$ to get the posterior probability $p(r_{\text{eff}}|L_{0.87}, L_{2.10})$ given measurements $L_{0.87}$ and $L_{2.10}$. Hereby, the arbitrary prior $p_{\text{pr}}(r_{\text{eff}})$ must be included within the bounds of the marginal probability $p_{\text{fwd}}(r_{\text{eff}})$ in the forward calculations. Values of $r_{\text{eff}}$ that are not included in
the forward calculations cannot be retrieved since the likelihood probability $p(L_{0.87}, L_{2.10}|r_{\text{eff}})$ is not defined for them. For given radiance measurements $L_{0.87}$ and $L_{2.10}$, Figure 1 shows an exemplary Monte Carlo approximation (blue histogram) of a posterior distribution (red line). With its mean and its standard deviation, the posterior distribution provides an estimation of the mean effective radius $\langle r_{\text{eff}} \rangle$.

## 2.3   Radiation transport model

The analysis of radiative transfer effects in one-dimensional clouds is done using *DISORT* (Stamnes et al., 1988). The representation of 3D radiative transfer in realistic cloud ensembles is done using the Monte Carlo approach with the *Monte Carlo code for the physically correct tracing of photons in cloudy atmospheres* (*MYSTIC*; Mayer,2009). In order to avoid confusion with the Monte Carlo sampling of posterior distributions mentioned above, this method will be termed "3D radiative transfer forward modeling" in the following. Both codes are embedded in the radiative transfer library *libRadtran* (Mayer
et al., 2005; Emde et al., 2016) which provides prerequisites and tools needed for the radiative transfer modeling. The atmospheric absorption is described by the representative wavelengths absorption parametrization (*REPTRAN*; Gasteiger et al., 2014). This parametrization is based on the HITRAN absorption database (Rothman et al., 2005) and provides spectral bands of different resolution ($1 \, \text{cm}^{-1}$, $5 \, \text{cm}^{-1}$, and $15 \, \text{cm}^{-1}$). Calculations have shown that the spectral resolution of $15 \, \text{cm}^{-1}$ (e.g., $\Delta\lambda = 1.1 \, \text{nm}$ at $870 \, \text{nm}$, $\Delta\lambda = 6.6 \, \text{nm}$ at $2100 \, \text{nm}$) best suits the spectral resolution of common hyperspectral imagers. The
extraterrestrial solar spectrum is based on data from Kurucz (1994) which is averaged over $1 \, \text{nm}$. In order to include vertical profiles of gaseous constituents, the standard summer mid-latitude profiles by Anderson et al. (1986) are used throughout this work. Since gaseous absorption is negligible at the chosen wavelength region of $(870.0 \pm 0.6) \, \text{nm}$ and $(2100.0 \pm 3.3) \, \text{nm}$, this choice still allows for a tropical as well as a mid-latitude application of the retrieval. Pre-computations of the cloud scattering phase function and single scattering albedo are done using the Mie tool *MIEV0* from Wiscombe (1980). When not mentioned

otherwise, a Gamma size distribution with $\alpha = 7$ was used for the Mie calculations. The high computational costs of the 3D Monte Carlo radiative transfer method for tracing large numbers of photons are reduced using the *Variance Reduction Optimal Option Method (VROOM)* (Buras and Mayer, 2011), a collection of various variance reduction techniques.

## 2.4 Cumulus cloud model

In order to calculate realistic posterior probability distributions $p(r_{\text{eff}}|L_{0.87}, L_{2.10})$, likelihood probabilities, produced by a sophisticated forward model, have to be combined with a realistic prior. While Marshak et al. (2006a) used statistical models to obtain this prior of 3D cloud fields, the physical consistency of cloud structures and cloud microphysics are an advantage of the explicit simulation of cloud dynamics and droplet interactions. Following Zinner et al. (2008), this work applies the three-dimensional radiative transfer model MYSTIC to realistic cloud fields which were generated with a *large eddy simulation (LES)* model on a cloud-resolving scale. While Zinner et al. (2008) uses realistic cloud structures combined with a bulk microphysics parametrization, this work extends their approach by including explicit simulations of entirely consistent, spectral cloud microphysics. In order to cover clean as well as polluted atmospheric environments, LES model outputs with different CCN concentrations will be used.

This work uses large-eddy simulations of Trade-wind cumulus clouds. The simulations were initially performed by Graham Feingold in the context of the Rain In Cumulus over Ocean (RICO) campaign (Rauber et al., 2007). The simulations use an adapted version of the Regional Atmospheric Modeling System (RAMS) coupled to a microphysical model (Feingold et al., 1996) and described in more detail in Jiang and Li (2009). In addition to the high spatial resolution, cloud microphysics are explicitly represented by size-resolved simulations of droplet growth within each grid box. The cloud droplet distributions cover radii between $1.56\text{-}2540\,\mu\text{m}$ which are divided into 33 size bins with mass doubling between bins. All warm cloud processes, such as collision-coalescence, sedimentation, and condensation/evaporation are handled by the method of moments developed by Tzivion et al. (1987, 1989). Droplet activation is included by using the calculated supersaturation field and a given cloud condensation nucleus concentration in two versions with $N_{\text{CCN}} = 100\,\text{cm}^{-3}$ and $N_{\text{CCN}} = 1000\,\text{cm}^{-3}$. The LES simulations (*dx25-100* and *dx25-1000*; Jiang and Li, 2009) have a domain size of $6.4 \times 6.4 \times 4\,\text{km}$ with a spatial resolution of $10\,\text{m}$ in the vertical and a spatial resolution of $25 \times 25\,\text{m}$ in the horizontal with periodic boundary conditions. As initial forcing, thermodynamic profiles collected during the RICO campaign (Rauber et al., 2007) were used. With condensation starting at a cloud base temperature of around $293\,\text{K}$ at $600\,\text{m}$, the cloud depth of the warm cumuli varies over a large range from $40\,\text{m}$ to a maximum of $1700\,\text{m}$ (Jiang and Li, 2009).

In order to sample a representative prior from this cumulus cloud simulations, a 2 hour ($12\,\text{h} - 14\,\text{h}$ LT) model output is sampled every $10\,\text{min}$ for both background CCN concentrations. As input for the following radiative transfer calculations, microphysical moments are derived from the simulated cloud droplet spectra. Using Equations (5) to (8), effective radius $r_{\text{eff}}$, liquid water content LWC and total cloud droplet concentration $N_d$ can be calculated from mass mixing ratios $m_i$ in $\text{g}\,\text{kg}^{-1}$

and cloud droplet mixing ratios $n_i$ in $\mathrm{kg}^{-1}$ given for the 33 LES size bins:

$$r_i(x,y,z) = \sqrt[3]{\frac{m_i(x,y,z)}{n_i(x,y,z)}\frac{3}{4\pi\rho_w}}, \tag{5}$$

$$r_{\mathrm{eff}}(x,y,z) = \frac{\sum_{i=1}^{33} r_i^3(x,y,z)\,n_i(x,y,z)\,\Delta r_i}{\sum_{i=1}^{33} r_i^2(x,y,z)\,n_i(x,y,z)\,\Delta r_i}, \tag{6}$$

$$\mathrm{LWC}(x,y,z) = \sum_{i=1}^{33} m_i(x,y,z)\rho_{\mathrm{air}}(x,y,z), \tag{7}$$

$$N_d(x,y,z) = \sum_{i=1}^{33} n_i(x,y,z)\rho_{\mathrm{air}}(x,y,z). \tag{8}$$

Figure 2 shows representative fields of cloud microphysics at $12\,\mathrm{h}\,40\,\mathrm{min}$ LT for the case with $N_{\mathrm{CCN}} = 1000\,\mathrm{cm}^{-3}$. In Figure 2A, the center panel (A1) shows a snapshot of the liquid water path (LWP). The side panel (A2) shows a north-south and (A3) a east-west cross-section of the liquid water content field. With $1067\,\mathrm{g\,m}^{-2}$, the LWP maximum is found co-located with a LWC maximum of over $2\,\mathrm{g\,m}^{-3}$ inside the strongest convective core. The inset in (A3) contains a zoomed view of the LWC gradient at cloud edge. For the same scene, Figure 2B provides an overview of optical thickness $\tau$ in the center panel (B1). The side panel (B2) shows a north-south and (B3) a east-west cross-section of the effective radius field. In the cross-sections of $r_{\mathrm{eff}}$, the growth of cloud droplets with height is visible. With an overall cloud fraction of $7.6\%$ and a mean optical thickness $\tau = 27$ for cloudy regions with $\mathrm{LWP} > 20\,\mathrm{g\,m}^{-2}$, the maximum optical thickness of $\tau_c = 176$ is found at the convective core as well. At the same time, the case with $N_{\mathrm{CCN}} = 100\,\mathrm{cm}^{-3}$ has a lower LWP maximum of $660\,\mathrm{g\,m}^{-2}$ and a lower mean cloud optical thickness of $\tau_c = 11$, while the cloud fraction is a little bit higher with $8.9\%$.

In the following, the variation of the vertical cloud droplet growth is explored in more detail since this is the main scientific objective of the proposed retrieval. Figure 3 shows contoured frequency by altitude diagrams (CFADs; Yuter and Houze, 1995) for $r_{\mathrm{eff}}$, LWC and total cloud droplet number concentration $N_d$. The black lines summarize the typical profiles of $r_{\mathrm{eff}}$, LWC and $N_d$ for $N_{\mathrm{CCN}} = 1000\,\mathrm{cm}^{-3}$, the red lines for $N_{\mathrm{CCN}} = 100\,\mathrm{cm}^{-3}$. The frequently occurring low values of $N_d$ and LWC are associated with grid boxes at cloud edges while the wide spectrum of larger values are located within the cloud cores. While effective radii sharply increase from $3\,\mu\mathrm{m}$ after droplet activation at cloud base to $12\,\mu\mathrm{m}$ (for $N_{\mathrm{CCN}} = 100\,\mathrm{cm}^{-3}$: $24\,\mu\mathrm{m}$) at cloud top ($h = 1.7\,\mathrm{km}$) with a small spread, the LWC increases gradually from cloud base to $0.9\,\mathrm{g\,m}^{-3}$ at $h = 1.5\,\mathrm{km}$ with a broad spread of LWC values. Above $1.5\,\mathrm{km}$, convection is capped by a subsidence inversion where cloud liquid water accumulates to values of up to $1.5\,\mathrm{g\,m}^{-3}$. As intended, the two cloud ensembles cover a wide range of possible values for $r_{\mathrm{eff}}$ and $N_d$ between low ("clean") and high CCN concentration ("polluted"). Small droplet $r_{\mathrm{eff}}$ and slow droplet growth with height characterize cases with $N_{\mathrm{CCN}} = 1000\,\mathrm{cm}^{-3}$, while fast droplet growth to larger $r_{\mathrm{eff}}$ values are a characteristic of the cases with $N_{\mathrm{CCN}} = 100\,\mathrm{cm}^{-3}$. LWC and cloud lower and upper boundaries show only small differences.

Based on the cloud base droplet number $N_{\mathrm{cb}}$, temperature $T_{\mathrm{cb}}$, pressure and a saturation adiabatic lapse rate (here we assume $4\,\mathrm{K\,km}^{-1}$), "adiabatic" reference values can be calculated for an ensemble of droplets growing by condensation during ascent, neglecting entertainment of dry environmental air (dashed lines). The existence of other effects (e.g., entrainment, coalescence) becomes evident in comparison with modeled LWC and $N_d$ profiles, as the adiabatic theory provides only an upper limit to

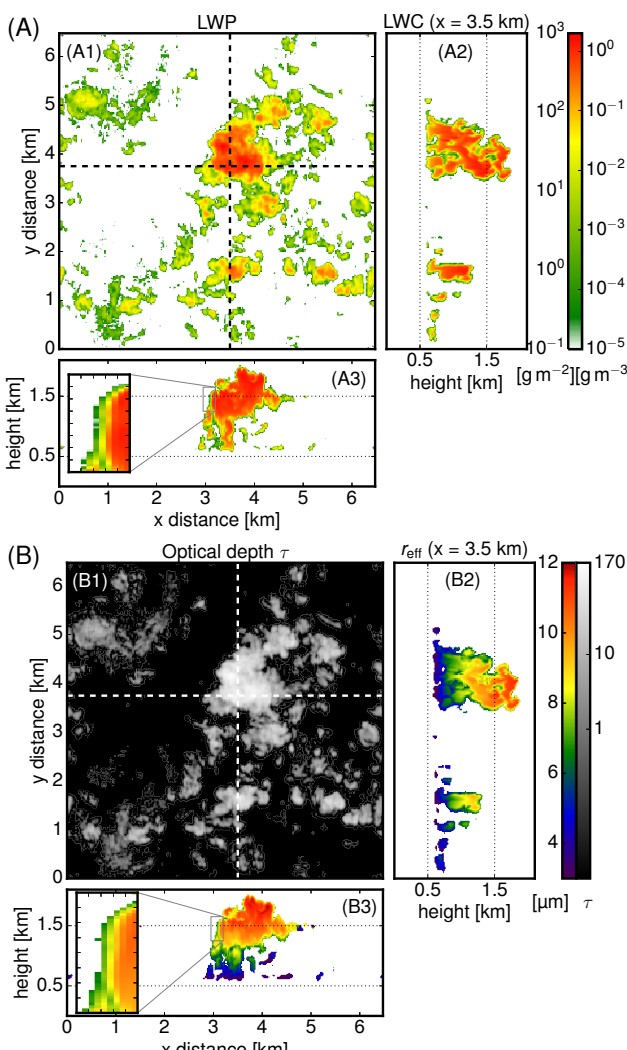

**Figure 2. (A)** Snapshot of LES cloud fields at $12\,\mathrm{h}\,40\,\mathrm{min}$ LT with **(A1)** liquid water path in $\mathrm{g\,m^{-2}}$ and **(A2)** north-south and **(A3)** east-west cross-sections of the liquid water content field in $\mathrm{g\,m^{-3}}$. **(B)** Same LES snapshot with **(B1)** optical thickness $\tau$ and **(B2)** north-south and **(B3)** east-west cross-sections of effective radius $r_{\mathrm{eff}}$ in $\mu$. The insets in A3 and B3 contain zoomed cross-sections of LWC and $r_{\mathrm{eff}}$ for a cloud edge region showing signs of lateral entrainment.

their values. In contrast, $r_{\mathrm{eff}}$ follows the adiabatic limit more closely with sub-adiabatic values between $60-80\%$ which is agrees with in-situ aircraft observations during the RICO campaign (Arabas et al., 2009) and other studies (Martin et al., 1994).

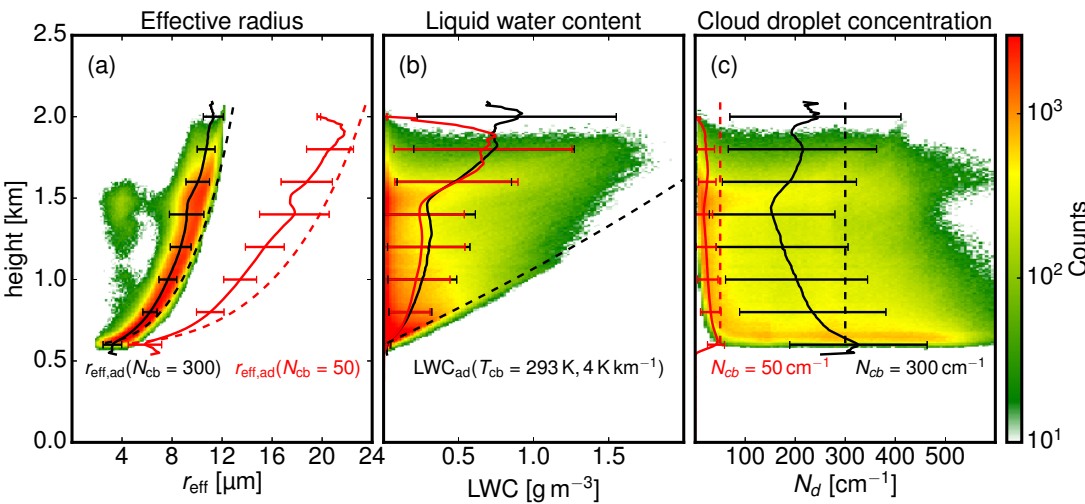

**Figure 3.** The contoured frequency by altitude diagrams (CFADs) show **(a)** effective radius $r_{\mathrm{eff}}$, **(b)** liquid water content LWC and **(c)** cloud droplet number concentration $N_d$ for the polluted cases with $N_{\mathrm{CCN}} = 1000\,\mathrm{cm}^{-3}$. Respective mean profile (black solid line) and its standard deviation (error bar) are superimposed. For the polluted cases with $N_{\mathrm{CCN}} = 100\,\mathrm{cm}^{-3}$, only the mean profiles are shown (red solid lines). In both cases, the dashed profile is the theoretical adiabatic limit calculated for conditions at cloud base ($T_{\mathrm{cb}} = 293\,\mathrm{K}, 4\,\mathrm{K\,km}^{-1}$) and $N_{\mathrm{cb}} = 300\,\mathrm{cm}^{-3}$ for the polluted and $N_{\mathrm{cb}} = 50\,\mathrm{cm}^{-3}$ for the clean case.

## 3 Methods

### 3.1 Selection of suitable cloud sides

A key component of the Bayesian approach is the selection of a suitable sampling strategy to explore the likelihood distribution $p_{\mathrm{fwd}}(L_{0.87}, L_{2.10} | r_{\mathrm{eff}})$. A suitable sampling strategy becomes furthermore essential when a computational expensive 3D
radiative transfer method is used to sample the observation parameter space. Following Mosegaard and Tarantola (1995), the sampling of the model space should always fit the expected measurement range. Instead of sampling the radiative transfer in 3D cloud fields at random, the indented measurement location and perspective should be taken into account.

To that end, we introduce a technique to select suitable locations within the LES model output for which cloud sides are visible from the airborne perspective. Cloud side measurements are intended for clouds within several kilometers from the
instrument location. With the sun in the back, azimuthal positions of $\pm 45°$ around the principal plane will be accepted for an airborne field of view, which is centered slightly below the horizon. In the following, an analytical method ensures the reproducibility to select observation locations. Here, an observation kernel $k_{\mathrm{FOV}}$ models the field-of-view with an azimuthal opening angle of $\Delta\varphi = 45°$ and a zenithal opening angle of $\Delta\vartheta = 40°$, centered around $5°$ below the horizon. As a function of radial distance, the observation kernel comprises a scalar weighting to curtail the desired location of clouds. In Figure 4a, a
three-dimensional visualization of the observation kernel method is presented. While the observation position (yellow dot) is moved through the model domain, the result of the convolution between observation kernel and cloud field is shown as arbitrary

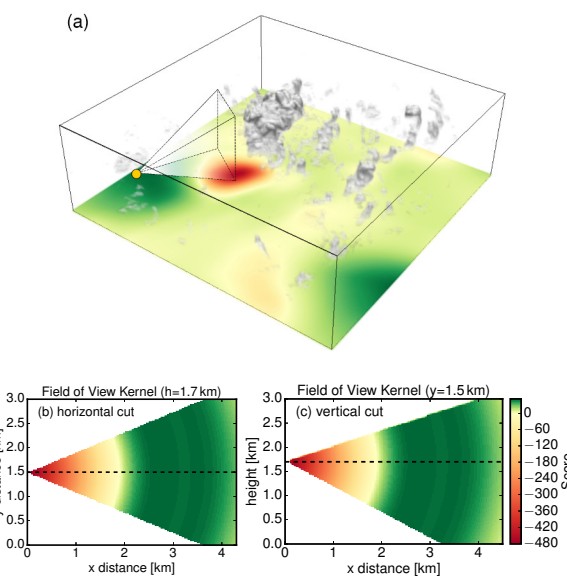

**Figure 4. (a)** Finding the optimal observation location for cloud side measurements. The surface shows the location score derived by convolving the observation kernel (shown in Figure 4b,c) with the LES cloud field (12 h 40 min LT) presented in Figure 2 **(b)** Horizontal and **(c)** vertical cross-sections of the observation kernel. The arbitrary score is positive for regions where clouds are desired.

score on the surface in Figure 4a. A more detailed view on the observation kernel is given in Figure 4b by a horizontal and in Figure 4c by a vertical cut at the dashed cutting line. The arbitrary score is strongly negative in the vicinity of the observer to penalize locations where clouds are too close. Observation distances of $3 \, \text{km}$ to $5 \, \text{km}$ turned out to maximize the likelihood to observe a complete cloud side in the used LES model output. For a distance of $2 \, \text{km}$ and onward, the weighting score becomes thus positive with a maximum at $3.5 \, \text{km}$ to favor locations with clouds in this region. For all LES cloud fields on average, this method positions the observer at a distance of around $4 \, \text{km}$ from cloud sides.

Subsequently, the field of cloudy grid boxes is convolved with the observation kernel at an observation altitude of $h = 1.7 \, \text{km}$, creating a two-dimensional score field $s_{\text{obs}}$. For every cloud field and chosen azimuthal orientation, the observation position is then placed where $s_{\text{obs}}$ has its global maximum. In Figure 4a, the already introduced LES cloud field (12 h 40 min LT) is shown in combination with the corresponding score field $s_{\text{obs}}$ obtained for a viewing azimuth of $\phi = 315°$. The yellow dot indicates the observation position, where $s_{\text{obs}}$ has its global maximum as recognizable by the green color. Also depicted is the field of view towards the largest cloud in the center of the domain. The red region in $s_{\text{obs}}$ would be unfavorable for a cloud side perspective since it would be too close to the cloud. For the selected perspective shown in Figure 4a, a simulated true-color image is shown in Figure 5.

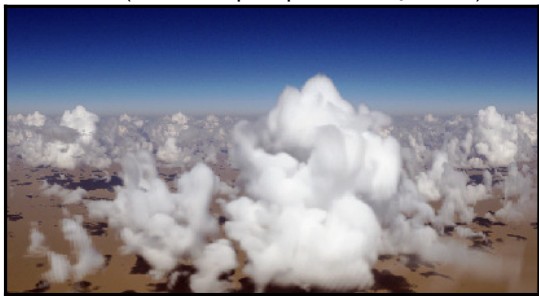

**Figure 5.** True-color image of a scene selected with the observation kernel method shown in Figure 4.

### 3.2 Determination of the apparent effective radius

As various studies have pointed out, the process of deriving the LES variables for $r_{\text{eff}}$ in the first place is not straightforward (Alexandrov et al., 2012; Miller et al., 2016; Zhang et al., 2017; Miller et al., 2018). First, $r_{\text{eff}}$ has to be derived from model parameters which describe the particle size distribution. This step was explained in Section 2.4 by Equations (5) to (8). Secondly,

an approach to infer the visible effective radius has to be developed in case of inhomogeneous cloud microphysics. In their statistical retrieval approach, Zinner et al. (2008) traced along the line of sight of each sensor pixel until hitting the first cloudy model grid box from which they selected their $r_{\text{eff}}$ corresponding to the observed radiances. This method has its limitations when it comes to highly structured cloud sides with horizontally inhomogeneous microphysics. The neglect of photon penetration depth disregards reflection from deeper within the cloud. In the solar spectrum, radiance observations contain information

from a multi-scattering path and not from the first grid box alone. With the line of sight method, the retrieved effective radius $r_{\text{eff}}$ becomes biased towards droplet sizes found directly at cloud edges. However, due to very low LWCs, these grid boxes have only a marginal contribution to the overall reflectance.

As Platnick (2000) showed, the penetration depth of reflected photons in the visible spectrum lies within some hundred meters while in the near-infrared spectrum the penetration depth is only a few dozen meters. The co-registration of responsible

cloud droplet sizes with modeled radiances is essential. Besides the observation perspective, this *apparent effective radius* $\langle r_{\text{eff}} \rangle_{app}$ also depends on the observed wavelength since different scattering and absorption coefficients lead to different cloud penetration depths.

In the following, a technique will be introduced to obtain $\langle r_{\text{eff}} \rangle_{app}$ during the Monte Carlo tracing of photons. As discussed by Platnick (2000), there exist analytical as well as statistical methods to consider the contribution of each cloud layer to the

apparent effective radius $\langle r_{\text{eff}} \rangle_{app}$. Advancing the one-dimensional weighting procedures of Platnick (2000) and Yang et al. (2003), the 3D tracing of photons in MYSTIC is utilized to calculate the optical properties of inhomogeneous, mixed-phase clouds. The apparent effective radius $\langle r_{\text{eff}} \rangle_{ph}$ for a photon is a weighted, linear combination of the individual effective radii

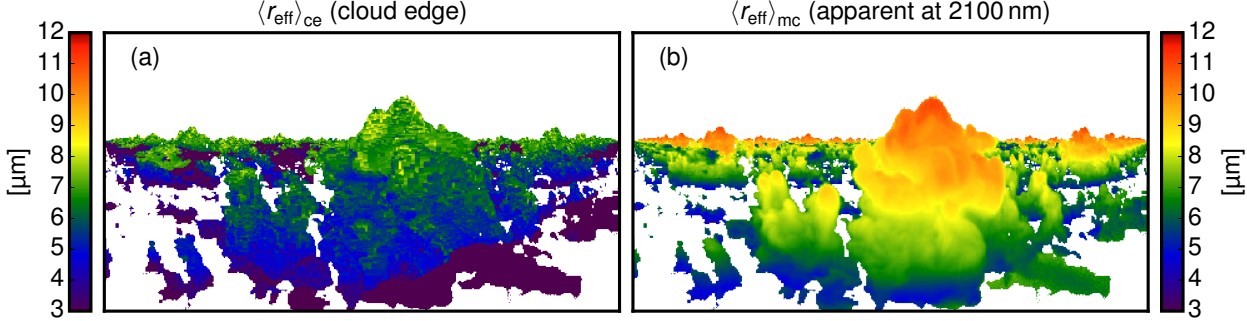

**Figure 6. (a)** Effective radii $r_{\text{eff}}$ found at cloud edge for the scene shown in Figure 5, **(b)** Apparent effective radii $\langle r_{\text{eff}} \rangle_{app}$ obtained with MYSTIC REFF for the same scene.

$r_{\text{eff}}$ the photon encounters on its path through the cloud:

$$\langle r_{\text{eff}} \rangle_{ph} = \frac{\int_0^l k_{ext}(l') \, r_{\text{eff}}(l') \, dl'}{\int_0^l k_{ext}(l') \, dl'}. \tag{9}$$

In Equation (9), the effective radii are weighted with the corresponding extinction coefficient $k_{ext}$ of the cloud droplets along the path length in each grid box. Subsequently, the mean over all photons traced for one forward simulated pixel leads to the apparent effective radius $\langle r_{\text{eff}} \rangle_{app}$ of this pixel:

$$\langle r_{\text{eff}} \rangle_{app} = \frac{\sum\limits_{i=0}^{photons} w_{ph,i} \, \langle r_{\text{eff}} \rangle_{ph,i}}{\sum\limits_{i=0}^{photons} w_{ph,i}}. \tag{10}$$

In Equation (10), the photon weight $w_{ph}$ describes the probability for the current photon path. With each absorption (or scattering) event, this weight is reduced until it reaches the detector where it is summed and converted into radiance. As the photon weights $w_{ph}$ are also used in the calculation of $L_{0.87}$ and $L_{2.10}$, the apparent effective radius $\langle r_{\text{eff}} \rangle_{app}$ can be derived simultaneously. This method was integrated within the MYSTIC 3D code and will, therefore, be referred to as the *MYStic method To Infer the Cloud droplet EFFective Radius (MYSTIC REFF)*.

For the cloud scene shown in Figure 5, Figure 6b shows the apparent effective radius $\langle r_{\text{eff}} \rangle_{app}$ obtained with MYSTIC REFF. Compared with the effective radius found at the cloud edge shown in Figure 6a, the apparent effective radius $\langle r_{\text{eff}} \rangle_{app}$ appears much smoother in Figure 6b. The range of values for $\langle r_{\text{eff}} \rangle_{app}$ obviously compares much better with the range of values of $r_{\text{eff}}$ shown in Figure 2(B3) and Figure 3a. The method shows very good agreement with the analytical solution of Yang et al. (2003) for homogeneous mixed-phase clouds and Platnick (2000) for one-dimensional clouds with a vertical effective radius profile.

## 4  The cloud geometry effect and its mitigation

Reflected radiance at non-absorbing wavelengths is mainly influenced by the optical thickness and by the amount of radiation incident on the cloud surface. For the latter, the cloud surface orientation relative to the sun is decisive. Therefore, an unknown cloud surface orientation is a challenge for all retrievals using radiances to derive $\tau_c$ and $r_{\mathrm{eff}}$ (e.g., Nakajima and King, 1990). In contrast to the typical observation geometry from above, where a plane-parallel cloud is assumed, the cloud surface orientation is mostly unknown for the cloud side perspective. In such a situation, where only the scattering angle $\vartheta_s$ is known, the limitation to optically thicker clouds can be a way out.

### 4.1  Limitation to optically thicker clouds

With increasing optical thickness $\tau_c$, the solar cloud reflectance becomes less sensitive to variations of $\tau_c$. This reduces an essential degree of freedom with respect to the radiative transfer. By "optically thicker", we refer to cumuli contained in the LES model output which exhibit well-developed cloud sides, e.g. like Cumuli mediocris, Cumuli congestus and Trade-wind cumuli. To give a concrete example, this term includes clouds with $\tau_c > 15$, e.g. with an average LWC of $0.5\,\mathrm{g\,m^{-3}}$, $r_{\mathrm{eff}} = 10\,\mu\mathrm{m}$ and with a vertical extent of $200\,\mathrm{m}$ and onward. Since the maximum optical thickness contained in the LES output is $\tau_c = 176$, the retrieval is designed for cumuli with $\tau_c = 15 - 150$. To dissect the impact of an unknown cloud surface orientation on the effective radius retrieval, the following study will use a "optically thick" water cloud ($\tau_c = 500$). We subsequently develop a method to exclude cloud shadows and to mitigate radiance ambiguities for the cumulus clouds contained in the LES ensemble using the obtained insights.

### 4.2  Ambiguities of reflected radiances

In the following study, the ambiguity caused by the unknown cloud surface orientation and the remaining sensitivity to the effective radius will be explored. For this idealized study, molecular absorption and scattering will be neglected. Figure 7 shows the basic geometry for cloud side remote sensing. Here, the cloud surface normal is $\hat{\mathbf{n}}$, the illumination vector from the sun is $\hat{\mathbf{s}}$ and the viewing vector towards the observer is $\hat{\mathbf{v}}$. The viewing zenith angle $\vartheta$ and the sun zenith angle $\vartheta_0$ are referenced in ground frame coordinates. Corresponding to these two angles, two additional angles exist which describe the inclination of $\hat{\mathbf{s}}$ and $\hat{\mathbf{v}}$ on the oriented cloud surface: the local illumination angle $\vartheta_0^*$ and the local viewing angle $\vartheta^*$ with respect to the cloud surface.

### 4.2.1  Principal plane (1D)

First, all vectors are assumed to be within the principal plane (the plane spanned by $\hat{\mathbf{s}}$ and $\hat{\mathbf{n}}$). Figure 7 shows two different viewing angles onto a vertical cloud surface and same local illumination angle $\vartheta_0^*$. In the following study, global illumination and viewing geometry remain constant while the cloud surface is rotated in a clockwise direction. By varying the cloud surface normal, this approach explores the ambiguity of $L_{0.87}$ and $L_{2.10}$ in the context of an unknown cloud surface orientation. For the direct backscattering geometry (Figure 7, left), the local viewing angle and the local illumination angle are always equal

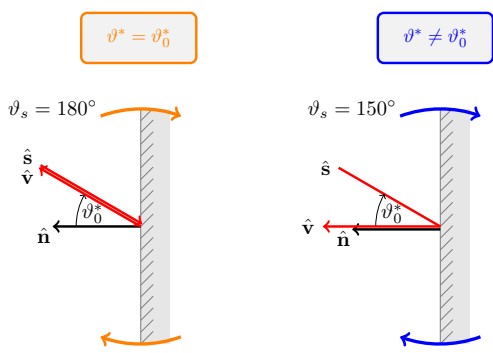

**Figure 7.** Two observation geometries with a scattering angle of $\vartheta_s = 180°$ **(left)** and $\vartheta_s = 150°$ **(right)**. Same cloud surface orientation $\hat{\mathbf{n}}$ (cloud surface normal) and illumination $\hat{\mathbf{s}}$ (solar direction vector) but different local viewing angle $\vartheta^*$. The impact of the indicated cloud surface rotation on reflected radiances is shown in Figure 8

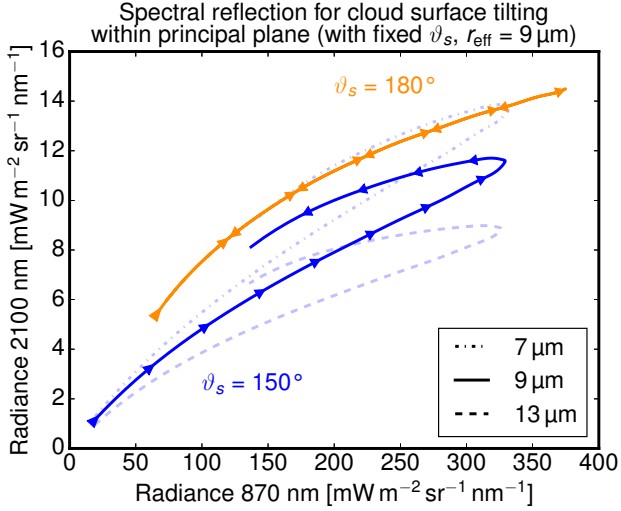

**Figure 8.** Spectral radiances used in 2-wavelengths retrievals at $\lambda = 870\,\text{nm}$ and $\lambda = 2100\,\text{nm}$ during the rotation of the cloud surface for observation geometries shown in Figure 7. Calculations of spectral reflection were done for an optically thick ($\tau = 500$) water cloud with a fixed effective radius $r_{\text{eff}} = 9\,\mu\text{m}$ with a fixed scattering angle of $\vartheta_s = 180°$ (orange line, $9\,\mu\text{m}$) and three different effective radii with a fixed scattering angle of $\vartheta_s = 150°$ (blue lines, $7\,\mu\text{m}$, $9\,\mu\text{m}$ and $13\,\mu\text{m}$).

($\vartheta^* = \vartheta_0^*$). In contrast, the local viewing angle can be larger ($\vartheta^* > \vartheta_0^*$) or smaller ($\vartheta^* < \vartheta_0^*$) than the local illumination angle for scattering angles $\vartheta_s < 180°$ (Figure 7, right).

For both cases, Figure 8 shows spectral radiances $L_{0.87}$ and $L_{2.10}$ during the clockwise rotation of the cloud surface. The radiative transfer calculations for this surface rotation of a water cloud were done with DISORT by varying the illumination and viewing angles while the scattering angle remained fixed. To exclude any effects of a varying optical thickness, the calculations

were done for a very high optical thickness of $\tau = 500$ and a fixed $r_{\mathrm{eff}} = 9\,\mu\mathrm{m}$. The arrows in Figure 8 indicate the progression of radiance values during the rotation of the cloud surface within the principal plane. The figure uses a typical two-channel diagram with the absorbing channel on the x-axis and the non-absorbing on the y-axis. E.g. Nakajima and King (1990) use this form to present the dependence of reflected radiance in both channels on the systematic variation of $\tau$ and $r_{\mathrm{eff}}$ values for plane-parallel clouds (hereafter denoted as "2-wavelength retrieval" and "2-wavelength diagram"). The similarity of these lines to the isolines for fixed $r_{\mathrm{eff}}$ and varying $\tau$ in their diagrams is striking.

Numerous studies (Cahalan et al., 1994; Varnai and Marshak, 2002; Zinner and Mayer, 2006; Vant-Hull et al., 2007) pointed out, that tilted and therefore more shadowed or illuminated cloud sides have a huge impact on the retrieval of optical thickness. The radiance similarity of cloud surface rotation and optical thickness variation further underlines the necessity to restrict the retrieval to optically thicker clouds (e.g. $\tau_c > 15$) when the cloud surface orientation is unknown. The following study will first focus on the "optically thick" water cloud ($\tau_c = 500$) to exclude any influence of optical thickness, In this way, the remaining information content for $r_{\mathrm{eff}}$ in $L_{0.87}$ and $L_{2.10}$ is determined.

### 4.2.2 Influence of scattering angle $\vartheta_s$

The obvious difference in Figure 8 between the direct backscatter case and the case with a scattering angle of $\vartheta_s = 150°$ highlights the influence of $\vartheta_s$ on the radiance ambiguity. While the radiance first increases at both wavelengths as the local illumination and viewing angle becomes smaller, it is only in case of direct backscatter that spectral radiances decrease the same way as they increased when the illumination angle becomes more oblique again. For $\vartheta_s = 150°$, spectral radiances $L_{2.10}$ are lower as long as $\vartheta^* < \vartheta_0^*$ when compared to the remaining part of the rotation when $\vartheta^* > \vartheta_0^*$.

As evident in Figure 8, a variable cloud surface orientation thus produces a characteristic bow structure outside of the direct backscatter geometry ($\vartheta_s < 180°$). For the optically thick cloud ($\tau_c = 500$) with unknown cloud surface orientation, this bow structure introduces ambiguity between $L_{0.87}$, $L_{2.10}$ and different effective radii. For an oblique viewing geometry ($\vartheta^* > \vartheta_0^*$), radiances from larger effective radii ($r_{\mathrm{eff}} = 9\,\mu\mathrm{m}$) coincide with radiances from smaller effective radii ($r_{\mathrm{eff}} = 7\,\mu\mathrm{m}$) for a steeper viewing geometry ($\vartheta^* < \vartheta_0^*$). For brighter cloud parts, however, there remain unambiguous regions where radiance pairs of different effective radii do not overlap.

### 4.2.3 Origin of ambiguity for $\vartheta_s < 180°$

For a deeper insight into the origin of the observed radiance ambiguity for $\vartheta_s < 180°$, we analyze the angular distribution of cloud reflectance at the absorbing and non-absorbing wavelength. In the following figures, the green dot will mark the cloud surface with a steeper local viewing angle (Figure 9, $\vartheta^* < \vartheta_0^*$) and the red dot the cloud surface with a more oblique local viewing angle (Figure 9, $\vartheta^* > \vartheta_0^*$). Contrary to the last study, the next Figure 10 shows radiances modeled for a fixed cloud surface orientation and illumination angle $\vartheta_0^* = 30°$ for different effective radii while the local viewing angle $\vartheta^*$ is varied.

Obviously, the angular characteristic differs between the absorbing and non-absorbing wavelength. Between the steep (green dot) and the oblique (red dot) viewing perspective, the radiance at the absorbing wavelength increases slightly while the radiance at the non-absorbing wavelength decreases. This asymmetric behavior becomes less pronounced for scattering angles

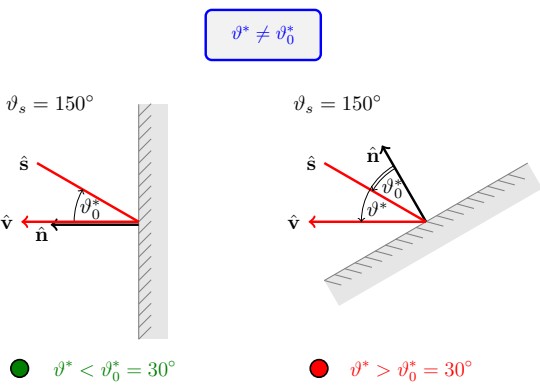

**Figure 9.** Two observation geometries with same scattering angle $\vartheta_s = 150°$ and same local illumination angle $\vartheta_0^* = 30°$. **(left)** Steep viewing direction perpendicular ($\vartheta^* = 0°$) to the cloud surface. **(right)** Oblique viewing perspective ($\vartheta^* = 60°$).

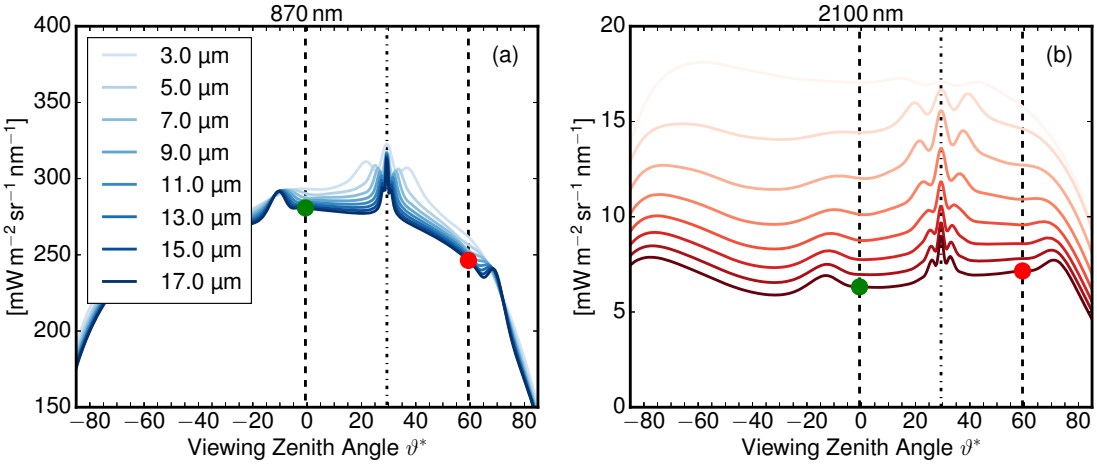

**Figure 10.** Spectral radiances at **(a)** $\lambda = 870\,\text{nm}$ and **(b)** $\lambda = 2.1\,\mu\text{m}$ for an optically thick water cloud ($\tau_c = 500$) for different effective radii as a function of relative viewing angle $\vartheta^*$ for a fixed illumination of $\vartheta_0^* = 30°$. The green and red dots mark viewing configurations shown in Figure 9 and Figures 11 and 12

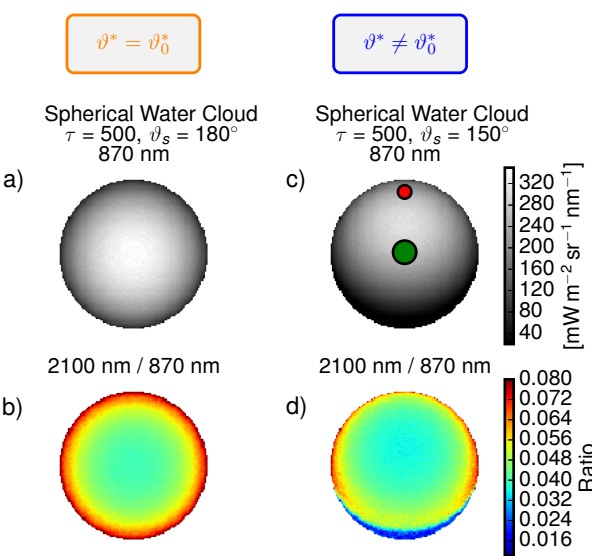

**Figure 11. (a,c)** Images of radiances $L_{0.87}$ and **(b,d)** of radiance ratios $L_{2.10}/L_{0.87}$ for the spherical and optically thick water cloud ($\tau_c =$ 500) with a fixed $r_{\mathrm{eff}} = 9\,\mu\mathrm{m}$ and for a fixed scattering angle of $\vartheta_s = 180°$ (left) and $\vartheta_s = 150°$ (right). The radiance ratio is shown to identify the origin of radiance pairs in the 2-wavelength diagram (Figure 12). As previously, the green and red dots mark viewing configurations shown in Figure 9.

near $\vartheta_s = 180°$. The reason for this different angular reflectance is connected with different photon penetration depths at the two wavelengths. The smaller penetration depth of near-infrared light leads to a more uniform reflection, while the larger penetration depth of visible light leads to a stronger reflection for the steep viewing perspective.

### 4.2.4 Spherical cloud (3D)

5  Next, the analysis is extended from principal plane considerations to a full 3D setup. To this end, 3D MYSTIC radiance simulations were done for a spherical, optically thick water cloud ($\tau_c = 500$) and different scattering regimes of $\vartheta_s = 180°$ and 150°. For the direct backscatter geometry ($\vartheta^* = \vartheta_0^*$) on the left and outside the direct backscatter geometry ($\vartheta^* \neq \vartheta_0^*$) on the right, Figure 11a,c show radiance images of $L_{0.87}$ and Figure 11b,d show radiance ratios $L_{2.10}/L_{0.87}$ for the spherical water cloud. The colored radiance ratios will later help to identify regions on the sphere within the 2-wavelength diagram. Furthermore, the two viewing geometries considered in Figure 9 are marked by the green and red dots.

10  Figure 12 shows the results in 2-wavelength diagrams for the direct backscatter direction (left) and for a scattering angle of 150° (right). In the 2-wavelength diagrams the radiance pairs from the 3D MYSTIC simulation are shown as scattered points, the results from the one-dimensional DISORT simulations for different effective radii are shown as black lines. Just like in Figures 9 to 11, the large green and red dots in Figure 12 indicate cloud surfaces with same local illumination angle $\vartheta_0^* = 30°$, but steeper ($\vartheta^* < \vartheta_0^*$, green dot) or more oblique local viewing angle ($\vartheta^* > \vartheta_0^*$, red dot).

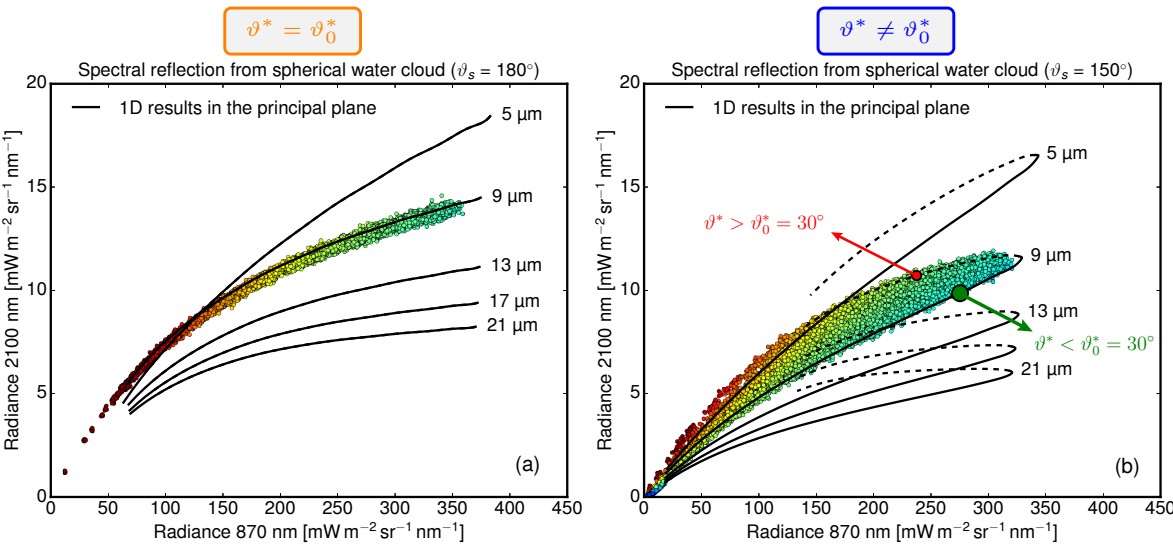

**Figure 12.** The scatter plots show the spectral radiances at $\lambda = 870\,\mathrm{nm}$ and $\lambda = 2.1\,\mathrm{\mu m}$ for the spherical cloud cases with a fixed $r_{\mathrm{eff}} = 9\,\mathrm{\mu m}$. **(a)** Results from Figure 11a for a fixed scattering angle of $\vartheta_s = 180°$ and **(b)** from Figure 11c for $\vartheta_s = 150°$. The color of the scatter points can be used to identify their location on the spherical water cloud in Figure 11b,d. For the same scattering angles and analogous to Figure 8, the black isolines show radiances from 1D DISORT simulations for an optically thick water cloud ($\tau_c = 500$) with different effective radii and variable cloud surface orientation. As previously, the green and red dots mark viewing configurations shown in Figure 9.

This ratio reflects the radial symmetry of the local illumination angle for $\vartheta_s = 180°$ (Figure 12a). For the direct backscatter geometry in Figure 12a, 3D results for $r_{\mathrm{eff}} = 9\,\mathrm{\mu m}$ match the 1D DISORT results for $r_{\mathrm{eff}} = 9\,\mathrm{\mu m}$ very closely. Due to the radial symmetry of the local illumination angles, radiance values also decrease radially symmetric with more oblique cloud surfaces. Albeit restricted to airborne or spaceborne platforms, this perspective minimizes the 3D effect on radiance ambiguities caused
5    by unknown cloud surface orientations. The picture changes when the observer leaves the backscatter geometry as shown for a scattering angle of $\vartheta_s = 150°$ in Figure 12b. As already shown with the DISORT results in Figure 8, the radiance pairs form a bow-like pattern with higher $L_{2.10}$ values at more oblique surface orientations. Furthermore, the red and green dots with same local illumination angle now become separated since $\vartheta^* \neq \vartheta_0^*$. While radiance at the non-absorbing wavelength drops considerably with a more oblique local viewing angle ($\vartheta^* > \vartheta_0^*$, red dot), radiance at the absorbing wavelength increase even
10    slightly. Consequently, droplets at the red dot with $r_{\mathrm{eff}} = 9\,\mathrm{\mu m}$ could be miss-interpreted as effective radius $r_{\mathrm{eff}} = 7\,\mathrm{\mu m}$ or even $5\,\mathrm{\mu m}$ .

Previous studies, like Marshak et al. (2006a) and Zinner et al. (2008), did not investigate in detail the origin of these radiance ambiguities. Nonetheless, they suggested to limit the influence of missing geometry information by additional consideration

of vertical thermal radiation temperature gradients (containing part of the geometry information). In the following a more systematic use of available geometry information in the visible and near-infrared spectrum is presented.

## 4.3 Additional information from surrounding pixels

Here, a technique is presented that uses information from surrounding pixels to resolve the radiance ambiguities caused by an unknown cloud surface orientation. Already Varnai and Marshak (2003) discussed and developed a method to determine how the surrounding of a cloud pixel influences the pixel brightness. In a recent study, Okamura et al. (2017) also used surrounding pixels to train a neural network to retrieve cloud optical properties more reliably. Our study will try to find a link between the pixel surrounding and the radiance ambiguity discussed in the preceding section.

As discussed in the preceding section, ambiguous radiances are caused by the asymmetric behavior of $L_{0.87}$ and $L_{2.10}$ when changing from a steep local viewing angle towards a more oblique perspective onto a cloud surface. While the pixel brightness $L_{0.87}$ decreases considerably at a more oblique perspective the pixel brightness $L_{2.10}$ increases. While changes in $L_{2.10}$ along a whole cloud profile are associated with a change in effective radius, the geometry-based brightness increase in $L_{2.10}$ should generally occur at smaller scales associated with cloud structures. The method should therefore determine if the surrounding pixels are darker or brighter at $\lambda = 2100\,\text{nm}$. At the same time, the method should be insensitive to instrument noise or Monte Carlo noise between adjacent pixels.

### 4.3.1 Comparison of pixel brightness

To this end, a 2D difference of Gaussians (DoG) filter is used to classify the viewing geometry onto the cloud surface in simulated as well as in measured radiance images. As a 2D difference filter, it compares the brightness of each pixel with the brightness of other pixels in the periphery. The filter consists of two 2D Gaussian functions $G_{\sigma_L}(x,y)$ and $G_{\sigma_H}(x,y)$ with different standard deviations $\sigma_L$ and $\sigma_H$ which specify the inner and outer search radius for the pixel brightness comparison:

$$G_{\sigma_H}(x,y) = \frac{1}{\sqrt{2\pi\sigma_H^2}} \exp\left(-\frac{x^2+y^2}{2\sigma_H^2}\right) \tag{11}$$

$$G_{\sigma_L}(x,y) = \frac{1}{\sqrt{2\pi\sigma_L^2}} \exp\left(-\frac{x^2+y^2}{2\sigma_L^2}\right) \tag{12}$$

Figure 13a shows the two Gaussians as a function of the angular distance from the considered pixel within the field of view. When the broader kernel $G_{\sigma_L}$ is subtracted from the narrower kernel $G_{\sigma_H}$ (Figure 13a, black line), the average pixel brightness within $\sigma_L$ is compared with the average pixel brightness of the center pixels within $\sigma_H$. This pixel brightness deviation $L_{2.10}^{DoG}$ is obtained by convolving the difference of $(G_{\sigma_H} - G_{\sigma_L})$ with the radiance image $L_{2.10}$:

$$L_{2.10}^{DoG}(x,y) = (G_{\sigma_H} - G_{\sigma_L}) * L_{2.10}(x,y) \tag{13}$$

Due to this subtraction, pixels are classified according to their positive or negative radiance deviation compared to their surrounding pixels. By using a not-too-small $\sigma_H$, not only the current pixel but a small surrounding is used, making the method less sensitive to noise of image sensors or Monte Carlo radiative transfer calculations.

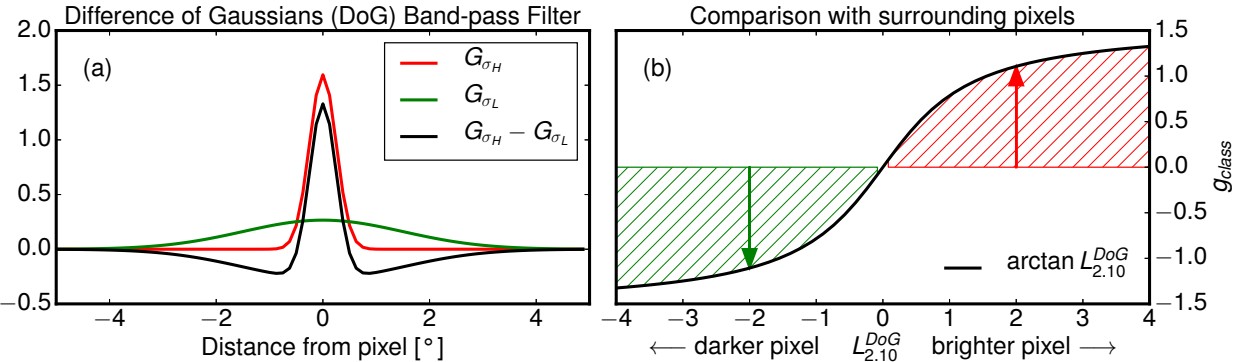

**Figure 13.** 2D Difference of Gaussians $G_{\sigma_H}$ (red) and $G_{\sigma_L}$ (green) which is used as a filter to derive The gradient classifier $g_{class}$ which compares the pixel brightness with the brightness of the surrounding pixels.

For the classification into steep or oblique perspectives, we are interested in the brightness deviation relative to the pixel surrounding. However, the absolute pixel brightness deviation $L_{2.10}^{DoG}$ can vary from scene to scene. To ease the binning of the gradient classifier $g_{\text{class}}(x,y)$, we constrain it into a fixed interval using the arctangent function:

$$g_{class}(x,y) = \arctan L_{2.10}^{DoG}(x,y) \tag{14}$$

This restriction of $g_{\text{class}}$ to the range $[-\pi/2, \pi/2]$ is shown in Figure 13b where positive values indicate pixels which are brighter at $\lambda = 2100\,\text{nm}$ compared to the brightness of their surrounding.

### 4.3.2 Pixel brightness deviation as a proxy of 3D effects

In practice, the radiance ambiguity cannot be directly derived from passive radiance measurements without a detailed knowledge of the cloud surface orientation. Hence, the following study will investigate if the gradient classifier $g_{\text{class}}$ can be used as a proxy to resolve the discussed radiance ambiguity. To demonstrate the method, the cloud field illustrated in Figure 6 was used again for radiance calculations, but with a fixed effective radius of $r_{\text{eff}} = 8\,\mu\text{m}$. For this fixed effective radius, the broad radiance distribution of $L_{0.87}$ and $L_{2.10}$ shown in Figure 14a is mainly caused by the different cloud surface orientations discussed in the previous section. In order to identify the regions leading to the upper part of the radiance scatter cloud in Figure 14a, an exponential function was fitted (black line) to the data points to determine the positive (red) or negative (green) deviation $\Delta L_{2.10}$ from the best fit (black line) for each radiance pair.

In the following, this deviation $\Delta L_{2.10}$ is taken as a reference for a perfect separation of 3D radiance ambiguities. It is important to mention, that this deviation $\Delta L_{2.10}$ can only be determined when the effective radius is already known. A method that would yield a similar separation without prior knowledge of $r_{\text{eff}}$ could be used as a proxy to mitigate the problem of ambiguous radiances. First, an optimal inner and outer search radius $\sigma_H$ and $\sigma_L$ has to be found to use $g_{\text{class}}$ as a proxy for the inaccessible radiance deviation $\Delta L_{2.10}$ This optimal search region was found by variation of search radii $\sigma_H$ and $\sigma_L$ and by subsequent correlation of the obtained $g_{\text{class}}$ in Figure 15b with the radiance deviation $\Delta L_{2.10}$ in Figure 15a. A maximum

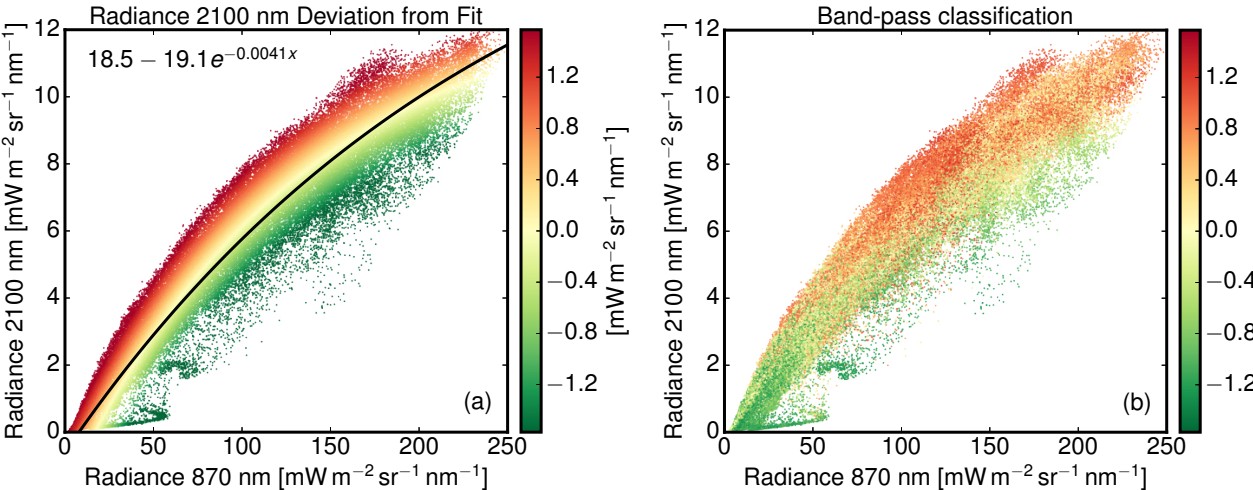

**Figure 14. (a)** 2-wavelength diagram for the MYSTIC calculation shown in Figure 5 but with a fixed effective radius of $r_{\text{eff}} = 8\,\mu\text{m}$. **(b)** Result of the gradient classifier $g_{\text{class}}$ applied to the same scene.

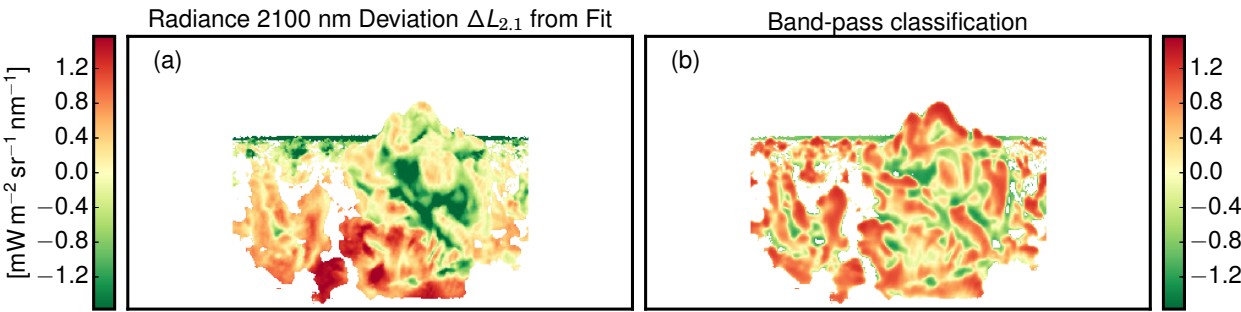

**Figure 15. (a)** Deviation $\Delta L_{2.10}$ from the fit in the 2-wavelength diagram in Figure 14a used as a reference for the **(b)** gradient classifier $g_{\text{class}}$ which puts the pixel radiance into context with surrounding pixels.

correlation with $\Delta L_{2.10}$ was found when the filter operated between $\sigma_H = 0.25°$ and $\sigma_L = 1.5°$. In spatial terms, the brightness within a search region of $30\,\text{m}$ to $150\,\text{m}$ is compared with the considered pixel brightness. For the optimal search radii, Figure 14 compares the radiance separation provided by the gradient classifier in Figure 14b with the reference in Figure 14a. Figure 15 shows reference and proxy also as images, where the radiance deviation $\Delta L_{2.10}$ is shown on the left in Figure 15a and the gradient classifier on the right in Figure 15b.

Apparently, the radiance separation by the gradient classifier $g_{\text{class}}$ is similar to the radiance deviation $\Delta L_{2.10}$. It is able to separate the radiance distribution into positive and negative radiance deviations $\Delta L_{2.10}$ at high as well as at low radiances. Large $g_{\text{class}}$ values are more likely to be associated with a more oblique local viewing angle ($\vartheta^* > \vartheta_0^*$), while smaller values are more likely to be associated with a more steep local viewing angle ($\vartheta^* < \vartheta_0^*$). For two pixels with same illumination angle, $g_{\text{class}} > 0$ marks thus the upper *radiance branch* in Figure 14b, while $g_{\text{class}} < 0$ marks the lower *radiance branch*. Based on this

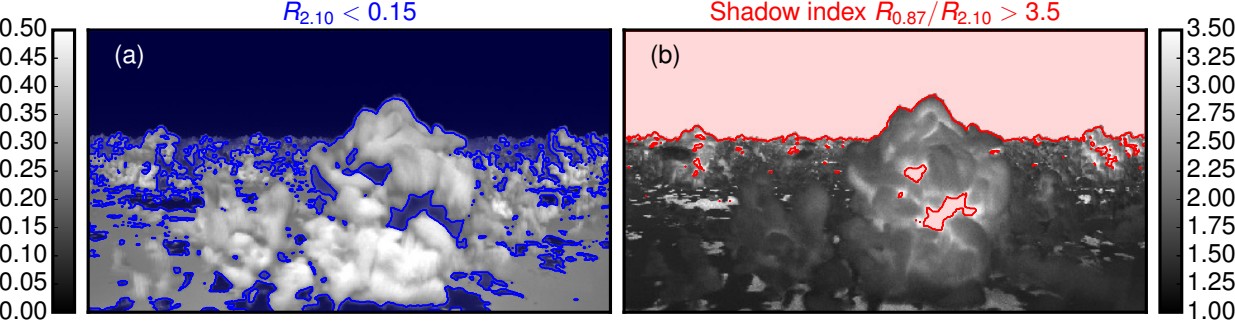

**Figure 16. (a)** Reflectivity at $2.1\,\mu$m for the cloud scene shown in Figure 6 (blue regions mark the simple reflectivity threshold $R_{2.10} < 0.15$). **(b)** Shadow index $R_{0.87}/R_{2.10}$ highlighting regions of enhanced cloud absorption caused by multiple diffuse reflections. (red regions mark the shadow index threshold $R_{0.87}/R_{2.10} > 3.5$)

feature, the gradient classifier $g_{\text{class}}$ can be used as a proxy to determine the geometry-induced radiance deviation of a pixel within the radiance distribution. For the retrieval, the gradient classifier $g_{\text{class}}$ is used for the 3D forward calculation ensemble as well as for real measurements.

## 4.4   Exclusion of cloud shadows

Cloud regions can also be self-shadowed if the local solar zenith angle onto the cloud surface $\vartheta_0'$ is larger than $90°$. Illuminated cloud parts can also cast shadows onto other cloud parts. Without direct illumination, reflected photons from these shadowed cloud parts originate from previous scattering events and are affected by those. For this reason, shadowed cloud parts have to be filtered out before applying any retrieval based on direct illumination.

Usually radiation from shadow regions encountered more absorption compared to directly reflected light (Vant-Hull et al., 2007). This enhanced absorption is visible in Figure 16a, where the reflectivity at $2.1\,\mu$m drops considerable for shadowed cloud regions. In Figure 16a, the blue areas illustrate a simple reflectivity threshold $R_{2.10} < 0.15$. As a proxy of enhanced absorption, the reflectivity ratio $R_{0.87}/R_{2.10}$ (Figure 16b) increases in this regions. In the following, this ratio will be used as *shadow index* $R_{0.87}/R_{2.10}$ to exclude pixels for which light has likely undergone multiple diffuse reflections:

$$R_{0.87}/R_{2.10} > 3.5 \qquad \text{(shadow index)} \tag{15}$$

In Figure 16b, the red areas marks regions with $R_{0.87}/R_{2.10} > 3.5$. The manual inspection of many cloud scenes confirmed $3.5$ as a viable shadow index threshold.

Unfortunately, clouds with very large cloud droplets ($r_{\text{eff}} > 12\,\mu$m) can exhibit similar high values of the shadow index. To study this limitation, DISORT calculations were done for an idealized water cloud to characterize the shadow index with respect to cloud optical thickness and effective radius. Figure 17 shows the shadow index as a function of effective radius $r_{\text{eff}}$ and optical thickness $\tau_c$ for a geometry ($\vartheta^* = 0°$, $\vartheta_0^* = 30°$) with high absorption at $2.1\,\mu$m. Like in Figure 16, the blue area indicates the simple reflectivity threshold $R_{2.10} < 0.15$ while the red area indicates the shadow index threshold $R_{0.87}/R_{2.10} >$

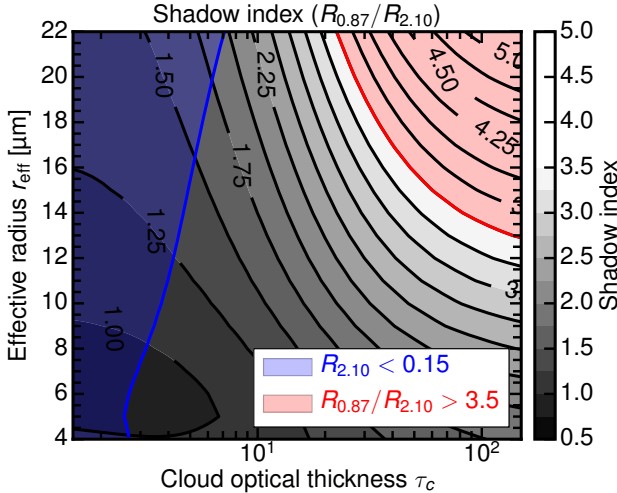

**Figure 17.** Shadow index $R_{0.87}/R_{2.10}$ for water clouds as a function of effective radius $r_{\mathrm{eff}}$ and cloud optical thickness $\tau_c$ for a geometry $(\vartheta^* = 0°, \vartheta_0^* = 30°)$ with high absorption at $2.1\,\mu m$. Like in Figure 16, the blue area indicates the simple reflectivity threshold $R_{2.10} < 0.15$ while the red area indicates the shadow index threshold $R_{0.87}/R_{2.10} > 3.5$.

3.5. Obviously, both shadow thresholds have their disadvantages. At higher optical thickness ($\tau_c > 100$), the shadow index $R_{0.87}/R_{2.10} > 3.5$ can confuse very large cloud droplets ($r_{\mathrm{eff}} > 12\,\mu m$) with cloud shadows. In contrast, the simple reflectivity threshold $R_{2.10} < 0.15$ can misidentify optical thin clouds ($\tau_c < 10$) as cloud shadows. The combined shadow mask $f_{\mathrm{shad}}$ of both thresholds in Equation (16) compensates the disadvantage of the shadow index threshold:

$$f_{\mathrm{shad}} = [R_{2.10} < 0.15 \quad \text{and} \quad R_{0.87}/R_{2.10} > 3.5] \qquad \text{(shadow mask)} \qquad (16)$$

In this way, only dark and highly absorptive cloud regions at $2.10\,\mathrm{nm}$ are classified as shadows. In addition, a threshold of $L_{0.87} > 75 \left[\mathrm{mWm}^{-2}\mathrm{nm}^{-1}\mathrm{sr}^{-1}\right]$ is used to focus the retrieval on optically thicker clouds and to filter out clear-sky regions.

## 5 Retrieval

In this section, the Monte Carlo sampled posterior distributions $p(r_{\mathrm{eff}}|L_{0.87}, L_{2.10})$ will be used to infer droplet size profiles from convective cloud sides. As mentioned in Section 2.2, the posterior $p(r_{\mathrm{eff}}|L_{0.87}, L_{2.10})$ can be derived from Bayes' theorem by solving the easier forward problem $p(L_{0.87}, L_{2.10}|r_{\mathrm{eff}})$ for all values of $r_{\mathrm{eff}}$.

The three-dimensional radiative transfer code MYSTIC is applied to LES model clouds to obtain simulations of realistic *specMACS* measurements. A whole ensemble of these MYSTIC forward simulations of cloud sides will then be incorporated within the statistical framework introduced in Section 2.1. Subsequently, the sampled statistics of reflected radiances are analyzed for their sensitivity to the effective cloud droplet radius.

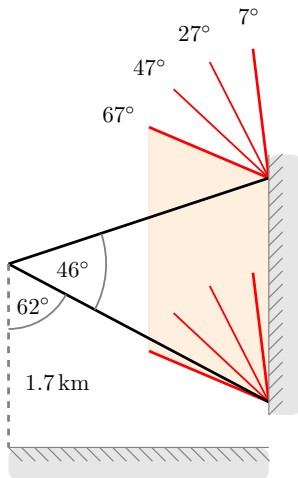

**Figure 18.** Setup of the viewing geometry ($\Delta\vartheta = 46°$, starting $62°$ from nadir) and the illumination geometry ($\vartheta_0 = 7, 27, 47$ and $67°$) for the airborne ($h = 1.7\,\text{km}$) 3D forward simulation ensemble. The horizontal extent of the field of view is $\Delta\varphi = \pm 45°$ from the principal plane.

## 5.1 Implementation of the 3D forward radiative transfer ensemble

In the following, an ensemble of 3D radiative transfer simulations is created to sample the posterior probability distribution $p(r_{\text{eff}}|L_{0.87}, L_{2.10})$. The ensemble of simulated cloud side measurements is set up by using the method to select suitable observation perspectives introduced in Section 3.1. During the radiative transfer calculations, the MYSTIC REFF method

(Section 3.2) determines the apparent effective radius which links the simulated radiances with the corresponding cloud droplet sizes. Despite the variance reduction methods in the MYSTIC code itself (Buras and Mayer, 2011), the time-consuming 3D technique still limits the number of model runs. Figure 18 illustrates the different illumination setups and the viewing geometry included within the 3D forward simulation ensemble. With LES cloud tops between $1.5$ and $2.0\,\text{km}$, the airborne perspective is set to an altitude of $h = 1.7\,\text{km}$. Since the retrieval should also be applicable in tropical regions, solar zenith angles were

chosen at $\vartheta_0 = 7, 27, 47$ and $67°$.

For each observation position selected in Section 3.1, images of cloud sides were simulated using MYSTIC. In line with the position selection method, the field of view of each image has an azimuthal opening angle of $\Delta\varphi = \pm 45°$, a zenithal opening angle of $\Delta\vartheta = 46°$ and is centered around $5°$ below the horizon. Comprising $720 \times 368$ pixels, each image was calculated with a spatial resolution of $0.125°$. For this image setup, solar radiances were calculated at the non-absorbing wavelength

$\lambda = 870\,\text{nm}$ ($L_{0.87}$) and the absorbing wavelength $\lambda = 2100\,\text{nm}$ ($L_{2.10}$). Since the width of the cloud droplet size distribution has no large impact on radiances at $L_{0.87}$ and $L_{2.10}$ (analysis not shown), the scattering properties were derived according to Mie theory using modified gamma size distributions with a fixed width of $\alpha = 7$ and the effective radius as simulated by

RAMS. For the ensemble, the surface albedo was set to zero since the influence of radiation reflected by vegetation on the ground is masked out in measurements. This technique will be described in the following part 2 paper.

Atmospheric aerosol was included by using the continental average mixture from the Optical Properties of Aerosols and Clouds (OPAC) package (Hess et al., 1998). The aerosol optical thickness (AOT) at $550\,\mathrm{nm}$ is around $\tau_a^{550} = 0.15$ for this profile. This aerosol profile is typical for anthropogenically influenced continental areas and contains soot and an increased amount of insoluble (e.g. soil) as well as water-soluble (e.g. sulfates, nitrates and organic) components.

A compromise had to be found to minimize the noise of the 3D Monte Carlo radiative transfer results and to keep computation time within reasonable limits. Here, the Monte Carlo noise should stay below the accuracy of the radiometric sensor which is assumed to be $\sim 5\%$. The photon number was thus chosen to be 2000 photons per pixel which leads to a standard deviation of about $2\%$.

All 12 RICO LES snapshots between $12\,\mathrm{h}\,00\,\mathrm{min}$ LT (local time) and $14\,\mathrm{h}\,00\,\mathrm{min}$ LT with a time step of $10\,\mathrm{min}$ were included in the 3D forward simulation ensemble. For four azimuth directions $\varphi = 45°, 135°, 225°$ and $315°$, suitable locations for cloud side observations were determined in each LES snapshot. For the polluted as well as for the clean cloud ensemble, $12 \times 4 = 48$ cloud scenes have been simulated for 4 solar zenith angles and 2 wavelengths with $720 \times 368$ pixels, totaling $101,744,640$ forward simulation pixels. In total, $2.0 \times 10^{11}$ photons have been traced on a computing cluster with 300 cores consuming $2.0 \times 10^8\,\mathrm{s}$ of CPU time.

## 5.2 Construction of the lookup table

In the next step, simulated radiances were binned into a multidimensional histogram with equidistant steps in $L_{0.87}$, $L_{2.10}$, $r_{\mathrm{eff}}$, $\vartheta$ and $g_{\mathrm{class}}$. Here, it is important to emphasize that the retrieval is designed to be independent from a-priori knowledge of $N_{\mathrm{CCN}}$. If the posterior distributions would separate between clean and polluted cases, the retrieval could tend to larger $r_{\mathrm{eff}}$ when a low $N_{\mathrm{CCN}}$ is measured. Such an retrieval would be unsuitable to study aerosol-cloud-interactions. For this reason, the radiance results from the polluted and the clean cloud ensemble are combined within the same histogram. Table 1 shows the specific binning of this histogram. During this discretization, radiances were counted in adjoining bins by linear interpolation. In the following, the histogram will be normalized to yield the posterior probability $p(r_{\mathrm{eff}}|L_{0.87}, L_{2.10})$.

## 5.3 Biased and unbiased priors

In the used cloud fields, the effective radius always increases with height which might impact the retrieval. The retrieval should not exhibit any trend towards a specific profile. Otherwise, the retrieval would reflect a-priori knowledge about the vertical profile of cloud microphysics. For this reason, the assumed prior is a key element to be considered in the sampling of the posterior and the subsequent Bayesian inference. In the case of cloud side remote sensing, two possible priors $p_{\mathrm{pr}}(r_{\mathrm{eff}})$ come into mind: a uniform prior or the LES model provided prior. For the LES model the prior is a function of viewing geometry, as some $r_{\mathrm{eff}}$ are more likely to be observed under certain viewing directions. In particular, relative frequency of $r_{\mathrm{eff}}$ for different scattering angles $\vartheta_s$ and gradient classes should be the same. For this problem, the prior probability should be uniform in $r_{\mathrm{eff}}$ to avoid the introduction of any bias.

**Table 1.** Variables, range and step size into which simulated radiances are binned to obtain a multidimensional histogram which is then used as a lookup table.

| Variable | Range | | Step | Bins |
|---|---|---|---|---|
| | Lower | Upper | | |
| $L_{0.87}^{*}$ | 0 | 290 | 5 | 58 |
| $L_{2.10}^{*}$ | 0 | 18 | 0.2 | 90 |
| $r_{\text{eff}}^{\dagger}$ | 3 | 25 | 2 | 11 |
| $\vartheta$ | $80°$ | $180°$ | $10°$ | 10 |
| $g_{\text{class}}$ | $-\frac{\pi}{2}$ | $+\frac{\pi}{2}$ | $\frac{\pi}{5}$ | 5 |

$*\,\mathrm{mWm}^{-2}\mathrm{nm}^{-1}\mathrm{sr}^{-1}$, $\dagger\,\mathrm{\mu m}$

Another important prerequisite of the Monte Carlo based Bayesian approach is the sufficient sampling of the likelihood probability $p(L_{0.87}, L_{2.10}, r_{\text{eff}})$. Naturally, effective radii not included in the ensemble of forward calculations cannot be retrieved using Bayesian inference. Furthermore, it should be kept in mind that sparsely sampled likelihood regions are probably not representative for the whole distribution. This is especially true for the smallest and largest effective radii contained in the LES model.

For this reasons an unbiased coverage of the likelihood probability is aspired. To this end, the ensemble with the *normal* cloud microphysics data from the LES model was complemented with calculations with vertically *flipped* cloud microphysics. The flipped cloud microphysics were derived by taking the additive inverse $-r_{\text{eff}}^{\text{orig}}$ of the original effective radius fields and add an offset $r_{\text{eff}}^{\text{offset}}$:

$$r_{\text{eff}}^{\text{flip}} = -r_{\text{eff}}^{\text{orig}} + r_{\text{eff}}^{\text{offset}}. \tag{17}$$

To ensure positive and realistic values for $r_{\text{eff}}^{\text{flip}}$, the offset $r_{\text{eff}}^{\text{offset}}$ was chosen to be at least $4\,\mathrm{\mu m}$ larger than the largest values found in all cloud fields. Thus, $r_{\text{eff}}^{\text{offset}} = 12\,\mathrm{\mu m} + 4\,\mathrm{\mu m} = 16\,\mathrm{\mu m}$ was used in Equation (17) for the polluted cloud ensemble $(\text{CCN} = 1000\,\mathrm{cm}^{-3})$ and $r_{\text{eff}}^{\text{offset}} = 22\,\mathrm{\mu m} + 4\,\mathrm{\mu m} = 26\,\mathrm{\mu m}$ for the clean cloud ensemble $(\text{CCN} = 100\,\mathrm{cm}^{-3})$. To preserve the optical thickness $\tau^{\text{orig}}$ of the original cloud field,

$$\tau^{\text{flip}} \equiv \tau^{\text{orig}}, \tag{18}$$

the well established relationship in Equation (19) was used to derive the liquid water content $\text{LWC}^{\text{flip}}$ for the flipped cases in Equation (20):

$$\tau \propto \frac{\text{LWC}}{r_{\text{eff}}}, \tag{19}$$

$$\text{LWC}^{\text{flip}} = \frac{r_{\text{eff}}^{\text{flip}}}{r_{\text{eff}}^{\text{orig}}} \text{LWC}^{\text{orig}}. \tag{20}$$

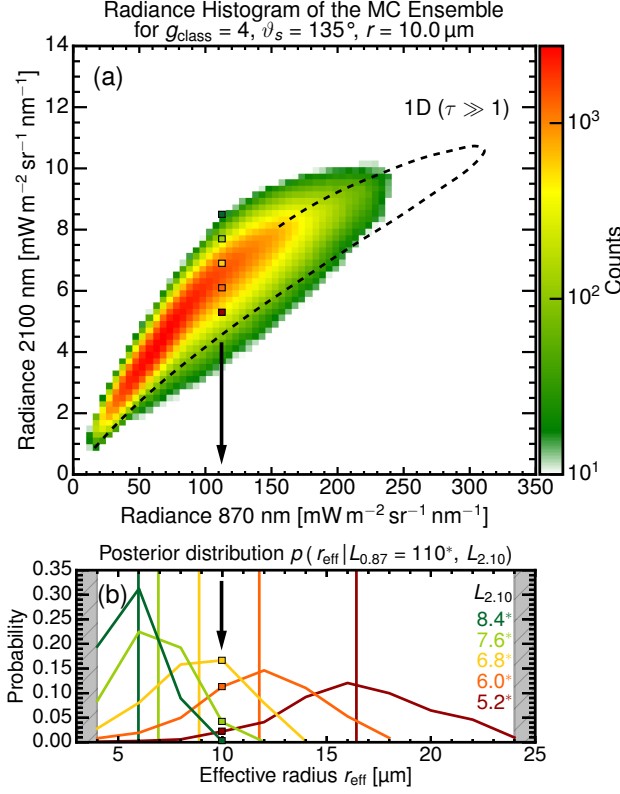

**Figure 19. (a)** Radiance histogram ($\vartheta_s = 135°$, $g_{\text{class}} = 4$) the 3D forward simulation ensemble of cloud sides which illustrates the radiance spread for the effective radius bin of $r_{\text{eff}} = 10\,\mu m$. The dashed line shows reflected radiances which were calculated with DISORT (1D RT code) for an optically thick ($\tau_c = 500$) water cloud for a variable cloud surface inclination within the principal plane. The colored dots indicate locations within the histogram for which the posterior distributions are shown in Figure 19b. **(b)** Corresponding posterior probability for a fixed radiance $L_{0.87} = 110\,\text{mWm}^{-2}\text{nm}^{-1}\text{sr}^{-1}$ at the non-absorbing wavelength and different $L_{2.10}$ radiances at the absorbing wavelength. The vertical lines indicate the corresponding mean effective radius for each posterior distribution.

## 5.4 Radiance and posterior distributions

The following section will present the radiance histograms $n(L_{0.87}, L_{2.10}, r_{\text{eff}})$ and the corresponding posterior distributions $p(r_{\text{eff}} | L_{0.87}, L_{2.10})$. Analogous to the likelihood distribution, the first gives the spread of radiances for a given effective radius $r_{\text{eff}}$, while the latter describes the spread of effective radii $r_{\text{eff}}$ for a given radiance pair $L_{0.87}$ and $L_{2.10}$. Figure 19a shows a 2D

5   histogram of simulated radiance combinations $L_{0.87}$ and $L_{2.10}$ for the airborne perspective. The histogram shows the results for the effective radius bin centered at $r_{\text{eff}} = 10\,\mu m$, the scattering angle bin between $\vartheta_s = 130°$ and $140°$ and the gradient class bin $g_{\text{class}} = 4$ which holds pixels that are brighter as their surroundings. The radiance spread from the three-dimensional model cloud sides, for the most part, can be explained by the one-dimensional DISORT results for $\tau_c = 500$ (dashed line for variable cloud surface inclination). After normalization of the histograms in Equation (3) and after the application of the

uniform prior in Equation (4), the posterior probabilities $p(r_{\mathrm{eff}}|L_{0.87}, L_{2.10})$ can be examined. Figure 19b shows posterior probabilities as a function of $r_{\mathrm{eff}}$ for different radiances $L_{2.10}$ at the absorbing wavelength corresponding to the colored dots in the histogram panel (Figure 19a). The vertical lines indicate the corresponding mean effective radius for each posterior distribution which were derived using Equation (21). The descending order of mean effective radii with ascending radiance $L_{2.10}$ demonstrates the general feasibility to discriminate different effective radii in cloud side measurements. Albeit a relatively large statistical retrieval uncertainty $\sigma(r_{\mathrm{eff}})$, the measurement of a radiance pair $(L_{0.87}, L_{2.10})$ can still narrow down $r_{\mathrm{eff}}$ to $\pm 1.5\,\mu\mathrm{m}$ around the most likely value. Interestingly, $\sigma(r_{\mathrm{eff}})$ increases with the effective radius from $\sigma(r_{\mathrm{eff}} = 6\,\mu\mathrm{m}) = \pm 1\,\mu\mathrm{m}$ to $\sigma(r_{\mathrm{eff}} = 16\,\mu\mathrm{m}) = \pm 3\,\mu\mathrm{m}$. In the following, the tabulated set of posterior distributions is used as lookup table for the effective radius retrieval.

## 5.5 Bayesian inference of the effective radius

Based on this lookup table of posterior probabilities $p(r_{\mathrm{eff}}|L_{0.87}, L_{2.10})$, the actual retrieval of effective radii can now be introduced. After a set of spectral radiance pairs $L_{0.87}$ and $L_{2.10}$ has been measured, the DoG filter (Section 4.3) is applied to the $L_{2.10}$ image to derive the gradient classifier $g_{\mathrm{class}}$. Scattering angles are calculated from the orientation and navigation data of the aircraft. With the four parameters, $L_{0.87}$, $L_{2.10}$, $g_{\mathrm{class}}$ and $\vartheta_s$ defined for each pixel, the corresponding posterior is retrieved from the lookup table by linear interpolation between posteriors defined at the bin centers of the lookup table. Finally, the mean effective radius $\langle r_{\mathrm{eff}} \rangle$ and the corresponding standard deviation $\sigma(r_{\mathrm{eff}})$ can be derived as first and second moments of the posterior distribution:

$$\langle r_{\mathrm{eff}} \rangle = \int r_{\mathrm{eff}}\, p(r_{\mathrm{eff}}|L_{0.87}, L_{2.10})\, dr_{\mathrm{eff}}. \tag{21}$$

$$\sigma(r_{\mathrm{eff}}) = \sqrt{\int \left(r_{\mathrm{eff}} - \langle r_{\mathrm{eff}} \rangle\right)^2 p(r_{\mathrm{eff}}|L_{0.87}, L_{2.10})\, dr_{\mathrm{eff}}}. \tag{22}$$

This 1-sigma standard deviation $\sigma(r_{\mathrm{eff}})$ will be referred to as the *statistical retrieval uncertainty*.

## 6 Numerical analysis of the retrieval

The next section will examine the stability of the statistical relationship between reflected radiance and cloud droplet size. How well can we retrieve the cloud droplet size after different viewing directions and cloud surface orientations have been combined within one lookup table? To answer this, the statistical retrieval is applied to simulated cloud side measurements for which the underlying effective radius is known. First, this is done for scenes that have already been included in the lookup table. Using scenes with normal and flipped effective radius profile, the lookup table is tested for an inherent bias towards a specific effective radius profile that could be caused by the chosen forward sampling strategy. In addition, tests are repeated for the same scenes with a fixed effective radius of $8\,\mu\mathrm{m}$; a case which is not included in the lookup table.

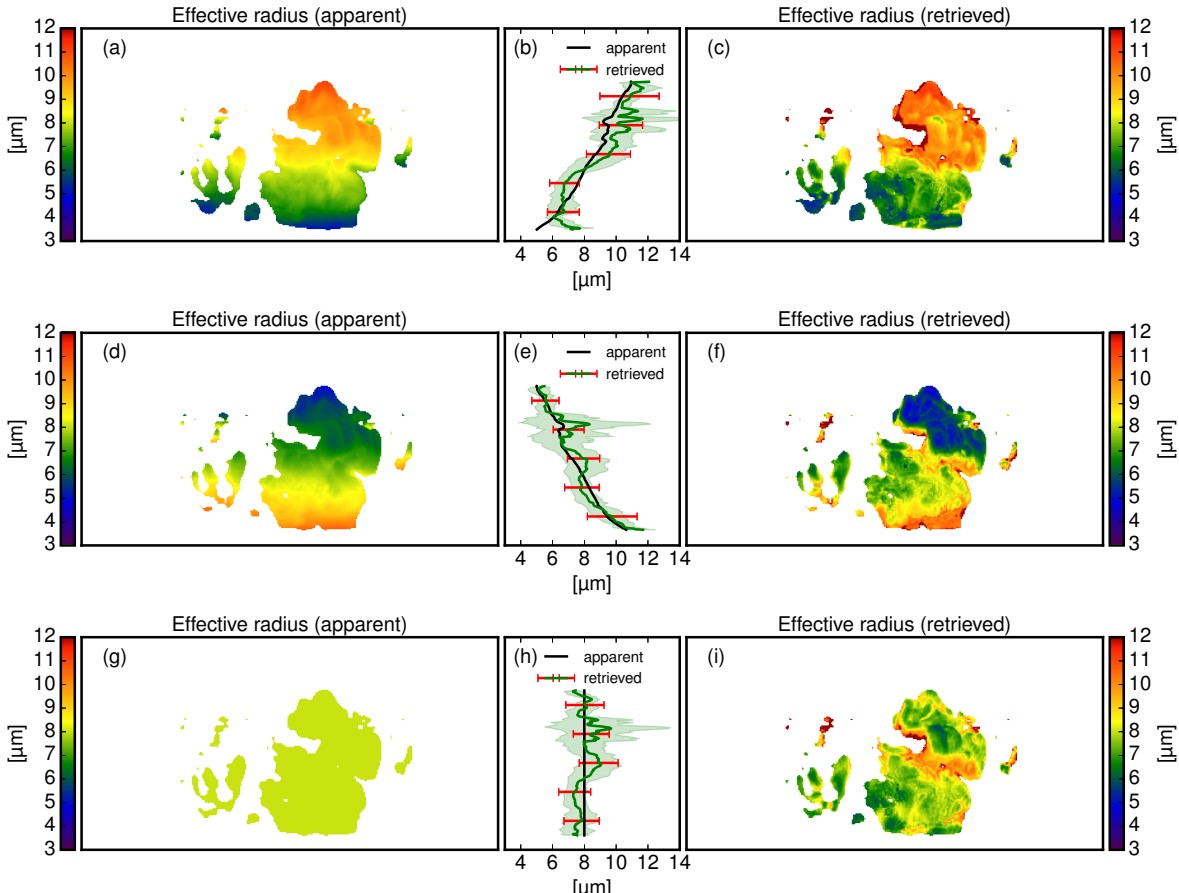

**Figure 20.** Retrieval test between the apparent effective radius (left) and the retrieved mean effective radius (right) for the normal (top), flipped (center) and fixed (bottom) effective radius profile. **(a)** Apparent effective radius $\langle r_{\mathrm{eff}} \rangle_{app}$ for the normal profile, **(b)** Mean and standard deviation of the apparent (black) and retrieved (green) vertical effective radius profile (normal) with the mean statistical retrieval uncertainty (red errorbars), **(c)** Retrieved mean effective radius for the normal profile, **(d)** Apparent effective radius $\langle r_{\mathrm{eff}} \rangle_{app}$ for the flipped profile, **(e)** Mean and standard deviation of the apparent (black) and retrieved (green) vertical effective radius profile (flipped) with the mean statistical retrieval uncertainty (red errorbars), **(f)** Retrieved mean effective radius for the flipped profile, **(g)** Apparent effective radius $\langle r_{\mathrm{eff}} \rangle_{app}$ for the fixed profile, **(h)** Mean and standard deviation of the apparent (black) and retrieved (green) vertical effective radius profile (fixed) with the mean statistical retrieval uncertainty (red errorbars), **(i)** Retrieved mean effective radius for the fixed profile

## 6.1 Analysis of the sampling bias

By design the retrieval should not exhibit any trend towards a specific profile. The polluted scene ($N_{\mathrm{CCN}} = 1000 \, \mathrm{cm}^{-3}$), already introduced in Figure 5, will be used as a first case study. Figure 20 shows result of the statistical effective radius

retrieval with the normal effective radius profile on top, the flipped profile in the center and a fixed effective radius profile at the bottom. The retrieved mean effective radius is shown in the right panels (Figure 20c,f,i), the apparent effective radius $\langle r_{\text{eff}} \rangle_{app}$ is shown in the left panels (Figure 20a,d,g). The center panels compare the mean vertical profile (lines) and its spatial standard deviation (shaded areas) of the apparent (black) and the retrieved (green) effective radius. Furthermore, the red error bars show the mean statistical retrieval uncertainty $\sigma(r_{\text{eff}})$ provided by the retrieval.

Like in Figure 19b, $\sigma(r_{\text{eff}})$ increases with $r_{\text{eff}}$ from $\sigma(r_{\text{eff}} = 6\,\mu\text{m}) = \pm 1\,\mu\text{m}$ to $\sigma(r_{\text{eff}} = 12\,\mu\text{m}) = \pm 2\,\mu\text{m}$. Within $\sigma(r_{\text{eff}})$, the retrieval reproduces the mean effective radius profile for all three cases quite well. For some specific cloud regions, however, there are also large differences of up to $\pm 3\,\mu\text{m}$. This is especially true at cloud edges and close to shadows.

Altogether, the statistical relationship between reflected radiance and cloud droplet size seems stable enough to be used for highly complex cloud sides. Moreover, these first results indicate that the retrieval seems to be resilient to the unrealistic, flipped cloud profiles included in its lookup table. Although the retrieval showed minor problems to retrieve the flipped profile, no substantial bias towards a specific effective radius profile could be detected. Mean values for all heights agree within the natural variability in the LES data; the retrieval error estimate $\sigma(r_{\text{eff}})$ seems to overestimate the uncertainty. Since these results were only obtained for a single cloud side scene, the following section will investigate these findings for a representative number of scenes.

**Statistic stability for included scenes**

In a first step, the retrieval will be tested for perspectives which are already included in the lookup table. This is done to test the retrieval for biases and to obtain a robust measure of correlation between the retrieval and the cloud side scenes it is composed of. By comparing this correlation with the correlation for cloud side scenes that are not included in the lookup table, this analysis will also be used to detect a potential over-fitting. There is the risk that the lookup table only reflects 3D effects that are specific for the included cloud side scenes.

In total, 9 cloud side perspectives were randomly chosen from the polluted as well as the clean dataset. For each perspective, the normal, the flipped as well as the fixed effective radius profile were tested. This amounts to $2 \times 9 \times 3 = 54$ test cases. For this statistical comparison, only reliable results with an retrieval error estimate $\sigma(r_{\text{eff}})$ of less than $2.5\,\mu\text{m}$ were included.

Figure 21a shows the correlation for the normal and the flipped polluted profiles which include around $358,000$ pixels. The linear regression with a slope of $0.97$ and an offset of $0.43\,\mu\text{m}$ shows no significant retrieval bias for these cases. The correlation between the apparent and the retrieved values is $0.78$. A deeper insight can be gained through Table 2, where all comparisons are summarized separately for normal and flipped profiles. The higher correlation coefficient ($0.80$ vs. $0.73$) seem to indicate a slightly better ability to detect the cases with a normal effective radius profile. Nevertheless, comparable linear regressions show no substantial bias towards the normal or the flipped cases. This confirms the observation made in the case study shown in Figure 20.

For the fixed effective radius profiles, the histogram in Figure 22 shows the deviation of the statistical retrieval with two distinct modes. With most likely values between $0\,\mu\text{m}$ and $-1\,\mu\text{m}$, the retrieval underestimates the effective radius slightly. A second mode is found where the retrieval overestimates $r_{\text{eff}}$ with values between $1\,\mu\text{m}$ and $2\,\mu\text{m}$. In combination, there is

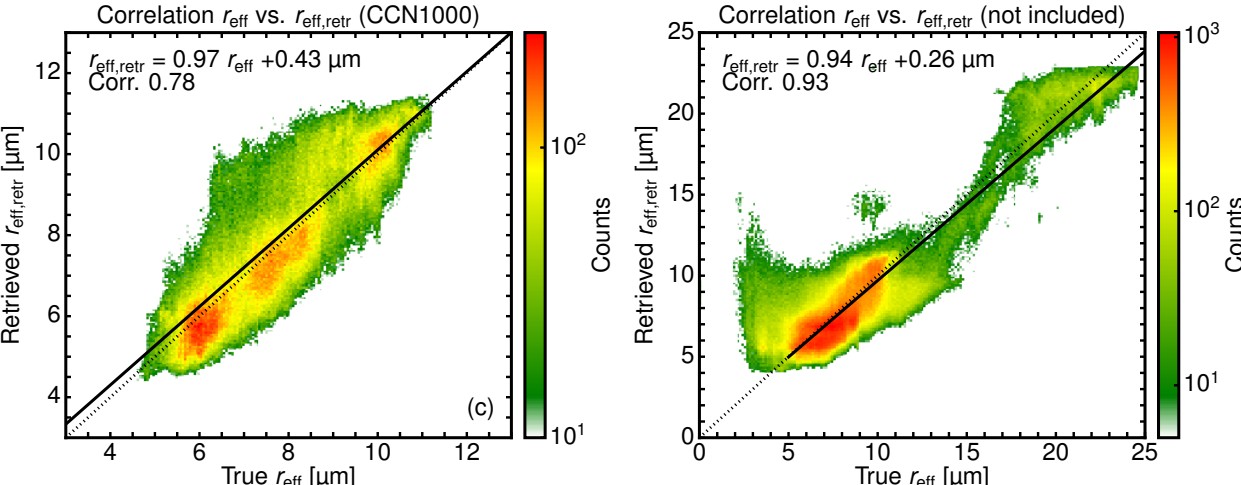

**Figure 21.** 2D histograms and linear regressions to determine the correlation between the apparent effective radius $r_{\text{eff}}$ and the retrieved effective radius $r_{\text{eff,retr}}$ for **(a)** the included polluted cases ($N_{\text{CCN}} = 100\,\text{cm}^{-3}$) with normal and flipped effective radius profile and **(b)** for the not included cases with normal and flipped effective radius profile.

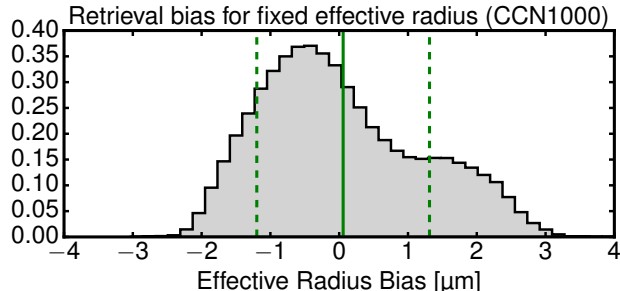

**Figure 22.** Retrieval deviations (retrieved $r_{\text{eff,retr}}$ - apparent $r_{\text{eff}}$) for the 9 cloud sides with fixed effective radius profile which are not included in the ensemble. The green line shows the average bias in retrieved effective radius $r_{\text{eff,retr}}$, the green dashed lines the root mean square error for $r_{\text{eff,retr}}$.

only a slight overestimation of $0.10\,\mu\text{m}$ with a larger standard deviation of $1.17\,\mu\text{m}$. A further investigation showed that the overestimation peak is connected with and found around undetected cloud shadows.

**Statistic stability for unknown scenes**

To check the retrieval for potential over-fitting, the retrieval was applied to unknown cloud side scenes. Nine new cloud side perspectives were selected from the polluted and the clean LES runs. While the forward ensemble contains viewing azimuths

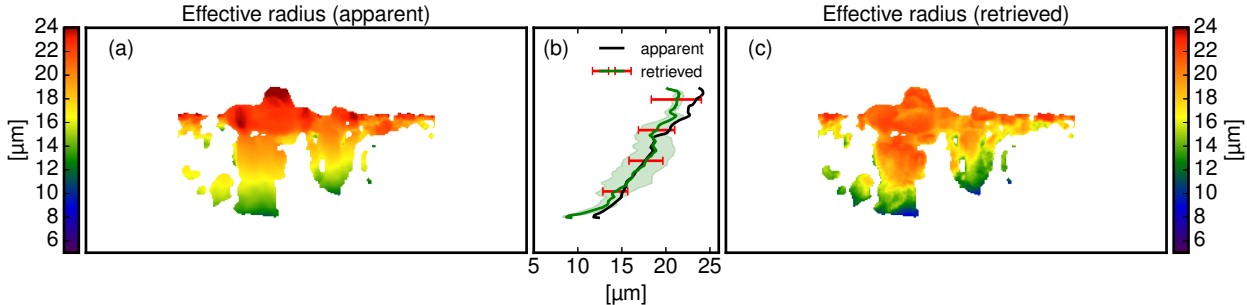

**Figure 23.** Retrieval comparison between the apparent effective radius (left) and the retrieved mean effective radius (right) for a cloud scene that was not included in the forward ensemble, **(a)** Apparent effective radius $\langle r_{\text{eff}} \rangle_{app}$ for the airborne perspective, **(b)** Mean and standard deviation of the apparent (black) and retrieved (green) vertical effective radius profile with the mean statistical retrieval uncertainty (red errorbars), **(c)** Retrieved mean effective radius for the airborne perspective.

of 45°, 135°, 225° and 315°, these new cloud side perspectives were chosen for new viewing azimuths of 0°, 90°, 180° and 270° and only normal effective radius profiles were used this time.

Figure 23 shows one of these new cloud side scenes with a normal effective radius profile. In contrast to Figure 20, this clean ($N_{\text{CCN}} = 100\,\text{cm}^{-3}$) scene features a much larger range of cloud droplet sizes. For this scene with overall larger $r_{\text{eff}}$, the retrieval underestimates $r_{\text{eff}}$ by $\pm 2.5\,\mu\text{m}$ at the upper cloud side part. Nevertheless, the statistical retrieval detects the mean effective radius profiles well within $\sigma(r_{\text{eff}})$. Like in Figure 20a, $\sigma(r_{\text{eff}})$ increases with $r_{\text{eff}}$ from $\sigma(r_{\text{eff}} = 12\,\mu\text{m}) = \pm 2\,\mu\text{m}$ to $\sigma(r_{\text{eff}} = 21\,\mu\text{m}) = \pm 3\,\mu\text{m}$.

The comparison for all not included cloud sides is shown in Figure 21b. The correlation for the not included cases is $0.93$, where around $339,000$ pixels are compared in total.

With nearly the same correlation coefficient, the retrieval performance remains the same when faced with unknown cloud side scenes. It can therefore be concluded that the retrieval is not trained only for the included cloud side scenes. Rather, it represents the statistical relationship between reflected radiance and cloud droplet size for this cloud ensemble.

## 7  Conclusions

The presented work advanced a framework for the remote sensing of cloud droplet effective radius profiles from cloud sides which was introduced by Marshak et al. (2006a); Zinner et al. (2008) and Martins et al. (2011). Up to now, their approach could not directly applied to realistic cloud side measurements (e.g. specMACS on HALO) since their studies lack the varying geometries of an airborne perspective. Furthermore, the effective radius was only parameterized and not directly calculated by a microphysical model. Moreover, Zinner et al. (2008) used the line of sight method to associate the forward modeled radiance with $r_{\text{eff}}$ found at the first cloudy grid box. To advance the technique to realistic airborne measurements, this study addressed following scientific objectives to overcome these limitations:

**Table 2.** Results of retrieval performance tests when faced with normal, flipped and not included effective radius profiles, grouped for $CCN = 1000$, $CCN = 100$ and for all not included cases. Linear regression, bias, RMSE and correlation are calculated between apparent and retrieved effective radius.

| Dataset | Slope | Offset | Bias | RMSE | Correlation |
|---|---|---|---|---|---|
| *CCN 1000* | | | | | |
| Included Profiles | +0.93 | +0.65 | -0.18 | +1.18 | +0.81 |
| - Normal Profiles | +0.95 | +0.49 | -0.12 | +1.20 | +0.80 |
| - Flipped Profiles | +0.96 | +0.52 | -0.24 | +1.17 | +0.73 |
| Unknown Profiles | +1.08 | -0.85 | +0.10 | +1.26 | +0.79 |
| *CCN 100* | | | | | |
| Included Profiles | +0.95 | +0.19 | -0.16 | +2.17 | +0.95 |
| Unknown Profiles | +1.05 | +0.25 | -0.26 | +2.28 | +0.78 |
| *All* | | | | | |
| Unknown Profiles | +0.94 | +0.26 | -0.01 | +1.86 | +0.93 |

1. Extend the existing approach to realistic airborne perspectives and develop methods to test the sensitivity of reflected radiances from cloud sides to cloud droplet radius, where the observer position is located within the cloud field. To this end, methods were developed to identify suitable observation positions within a model cloud field and to calculate an apparent effective radius for each forward modeled sensor pixel.

2. In this course, 3D radiative effects caused by the unknown cloud surface orientation were investigated. A technique was proposed to mitigate their impact on cloud droplet size retrievals by putting pixel in context with their surrounding.

3. Finally, an effective radius retrieval for the cloud side perspective was developed and tested for cloud scenes which were used during the retrieval development, as well as for unknown scenes.

The scope of this work was limited to the liquid part of convective water clouds, e.g. Cumulus mediocris, Cumulus congestus and Trade-wind cumulus, which exhibit well-developed cloud sides. In principle, the proposed technique could also be extended to ice clouds.

In a first step, this work introduced a statistical framework for the proposed remote sensing of cloud sides following Marshak et al. (2006a). A statistical relationship between reflected sunlight in a near-visible and near-infrared wavelength and droplet size is found following the classical approach by (Nakajima and King, 1990). By simulating the three-dimensional radiative transfer for high-resolution LES model clouds using the 3D Monte Carlo radiative transfer model MYSTIC, probability distributions for this relationship were sampled. These distributions describe the probability to find a specific droplet size after a specific solar reflectance pair of values has been measured. In contrast to many other effective radius retrievals, this work

thereby provides essential information about the retrieval uncertainties which are intrinsically linked with the reflectance ambiguities caused by three-dimensional radiative effects. Furthermore, this work developed a technique (Section 4.3) to reduce 3D radiance ambiguities when no information about the cloud surface orientation is available. More precisely, additional information from surrounding pixels was used to classify the environment of the considered pixel. Subsequently, this technique and the forward simulated probability distributions were incorporated into a statistical retrieval of the effective radius from cloud sides. Defined by the used LES model fields and the chosen geometries of the forward simulations, this retrieval is designed for

- cloud side measurements of the liquid part of convective water clouds, e.g., Cumulus mediocris, Cumulus congestus and Trade-wind cumulus, which exhibit well-developed cloud sides.

- cloud tops between $1.5 - 2\,\mathrm{km}$, an optical thickness between $15 - 150$ and effective radii between $4 - 24\,\mu\mathrm{m}$,

- spatially highly resolved ($10 \times 10\,\mathrm{m}$) images of the spectral radiance at $\lambda = 870\,\mathrm{nm}$ and $\lambda = 2100\,\mathrm{nm}$,

- a variable and unknown CCN background concentration between $100 - 1000\,\mathrm{cm}^{-3}$,

- variable sun zenith angles $\vartheta_0$ between $7 - 67°$ for tropical as well as mid-latitude application,

- an airborne perspective at $1.7\,\mathrm{km}$ altitude with a field of view of $\Delta\varphi = 46°$ (azimuthal) and $\Delta\vartheta = 40°$ (horizontal) centered $5°$ below the horizon.

The numerical analysis of the statistical retrieval showed a RMSE between retrieved and apparent $r_{\mathrm{eff}}$ of around 1-1.5 $\mu\mathrm{m}$. For the airborne measurement perspective aimed for (see part 2 of this work), the statistical retrieval reliably detects the present effective radius profile, while sanity checks showed no prior bias of the retrieval towards specific cloud droplet size profiles. This is an essential prerequisite for all consecutive interpretation of the retrieval results. Furthermore, the retrieval performance remained the same when faced with unknown cloud side scenes not included in the ensemble used for the retrieval. It can therefore be concluded that the retrieval is not over-fitted and that it represents the statistical relationship between reflected radiance and cloud droplet size for this cloud side perspective.

Moreover, this work dissected the impact of an unknown cloud surface orientation on bi-spectral effective radius retrievals using an optically thick water cloud sphere ($\tau_c = 500$). Just with every other 3D cloud side scene, some local viewing angles onto cloud surfaces are steeper while some viewing perspectives onto cloud surfaces are more oblique than the local illumination angle. As a consequence, the correlation between reflected solar radiance pairs and droplet sizes becomes ambiguous.

This study did not address the open question if and how strong rain can influence the retrieval performance. Several studies (Nakajima et al., 2009; Zinner et al., 2010; Zhang et al., 2012) found only a small impact of drizzle on bi-spectral methods of about $0.5 - 2\,\mu\mathrm{m}$. Although beyond the scope of this work, subsequent studies should address the influence of rain on bi-spectral retrievals for stronger precipitation rates as suggested by the more recent study of Zhang (2013).

The next important step is the application of the proposed retrieval technique to real measurements. In combination with simultaneous in-situ measurements, airborne cloud side observations have been acquired with the hyperspectral cloud and sky imager specMACS. In a follow-up paper part 2, the proposed retrieval will be validated with this independent in-situ data.

A further important point is the development of a distance mapping for the retrieval. The height and location assignment of retrieval results is not just of uttermost importance for the comparison with in situ measurements and models, but also essential to estimate the cloud distance for a potential aerosol correction. Here, first promising results could be achieved by exploiting the oxygen A-band absorption at $\lambda = 762\,\mathrm{nm}$ presented in Zinner et al. (2018). In conclusion, the present work developed a working effective radius retrieval to measurements of clouds sides applicable to real measurements and thus paved the way for further research on this topic.

*Acknowledgements.* We would like to thank Graham Feingold for providing the RICO cloud model fields. Thanks and acknowledgment also go to Silke Gross and the DLR Institute of Atmospheric Physics who supported the completion of this manuscript. Florian Ewald was partly funded by the German Research Foundation (DFG) under grant number MA 2548/9-1. Tobias Kölling was supported by the German Research Foundation (DFG) under grant number Zi 1132/3-1.

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

Wendisch, Manfred and Pöschl, Ulrich and Andreae, Meinrat O. and Machado, Luiz A. T. and Albrecht, Rachel and Schlager, Hans and Rosenfeld, Daniel and Martin, Scot T. and Abdelmonem, Ahmed and Afchine, Armin and Araùjo, Alessandro C. and Artaxo, Paulo and Aufmhoff, Heinfried and Barbosa, Henrique M. J. and Borrmann, Stephan and Braga, Ramon and Buchholz, Bernhard and Cecchini,

Micael Amore and Costa, Anja and Curtius, Joachim and Dollner, Maximilian and Dorf, Marcel and Dreiling, Volker and Ebert, Volker and Ehrlich, André and Ewald, Florian and Fisch, Gilberto and Fix, Andreas and Frank, Fabian and Fütterer, Daniel and Heckl, Christopher and Heidelberg, Fabian and Hüneke, Tilman and Jäkel, Evelyn and Järvinen, Emma and Jurkat, Tina and Kanter, Sandra and Kästner, Udo and Kenntner, Mareike and Kesselmeier, Jürgen and Klimach, Thomas and Knecht, Matthias and Kohl, Rebecca and Kölling, Tobias and Krämer, Martina and Krüger, Mira and Krisna, Trismono Candra and Lavric, Jost V. and Longo, Karla and Mahnke, Christoph and

Manzi, Antonio O. and Mayer, Bernhard and Mertes, Stephan and Minikin, Andreas and Molleker, Sergej and Münch, Steffen and Nillius, Björn and Pfeilsticker, Klaus and Pöhlker, Christopher and Roiger, Anke and Rose, Diana and Rosenow, Dagmar and Sauer, Daniel and Schnaiter, Martin and Schneider, Johannes and Schulz, Christiane and de Souza, Rodrigo A. F. and Spanu, Antonio and Stock, Paul and Vila, Daniel and Voigt, Christiane and Walser, Adrian and Walter, David and Weigel, Ralf and Weinzierl, Bernadett and Werner, Frank and Yamasoe, Marcia A. and Ziereis, Helmut and Zinner, Tobias and Zöger, Martin: ACRIDICON–CHUVA Campaign: Studying Tropical

Deep Convective Clouds and Precipitation over Amazonia Using the New German Research Aircraft HALO, Bulletin of the American Meteorological Society, 18, 1885–1908, https://doi.org/10.1175/BAMS-D-14-00255.1, 2016.

Wiscombe, W.: Improved Mie scattering algorithms, Applied Optics, 19, 1505–1509, 1980.

Yang, P., Wei, H., Baum, B., Huang, H., Heymsfield, A., Hu, Y., Gao, B., and Turner, D.: The spectral signature of mixed-phase clouds composed of non-spherical ice crystals and spherical liquid droplets in the terrestrial window region, Journal of Quantitative Spectroscopy

and Radiative Transfer, 79, 1171–1188, 2003.

Yuter, S. E. and Houze, R. A.: Three-Dimensional Kinematic and Microphysical Evolution of Florida Cumulonimbus. Part II: Frequency Distributions of Vertical Velocity, Reflectivity, and Differential Reflectivity, Monthly Weather Review, 123, 1941–1963, https://doi.org/10.1175/1520-0493(1995)123<1941:TDKAME>2.0.CO;2, http://journals.ametsoc.org/doi/abs/10.1175/1520-0493(1995)123%3C1941%3ATDKAME%3E2.0.CO%3B2, 1995.

Zhang, Z., Ackerman, A. S., Feingold, G., Platnick, S., Pincus, R., and Xue, H.: Effects of cloud horizontal inhomogeneity and drizzle on remote sensing of cloud droplet effective radius: Case studies based on large-eddy simulations, Journal of Geophysical Research: Atmospheres, 117(D19), 2012.

Zhang, Z.: On the sensitivity of cloud effective radius retrieval based on spectral method to bi-modal droplet size distribution: A semi-analytical model, Journal of Quantitative Spectroscopy and Radiative Transfer, 129(), 79–88, 2013.

Zhang, Z., Dong, X., Xi, B., Song, H., Ma, P. L., Ghan, S. J., Plat- nick, S., and Minnis, P.: Intercomparisons of marine boundary layer cloud properties from the ARM CAP-MBL campaign and two MODIS cloud products, J. Geophys. Res., 122, 2351–2365, https://doi.org/10.1002/2016JD025763, 2017.

Zinner, T. and Mayer, B.: Remote sensing of stratocumulus clouds: Uncertainties and biases due to inhomogeneity, Journal of Geophysical Research, 111, D14 209, https://doi.org/10.1029/2005JD006955, 2006.

Zinner, T., Marshak, A., Lang, S., Martins, J. V., and Mayer, B.: Remote sensing of cloud sides of deep convection: towards a three-dimensional retrieval of cloud particle size profiles, Atmos. Chem. Phys., 8, 4741–4757, 2008.

Zinner, T., Wind, G., and Platnick, S.: Testing remote sensing on artificial observations: impact of drizzle and 3-D cloud structure on effective radius retrievals, Atmos. Chem. Phys, 10, 9535-9549, 2010.

Zinner, T., Schwarz, U., Kölling, T., Ewald, F., Jäkel, E., Mayer, B., and Wendisch, M.: Cloud geometry from oxygen-A band observations through an aircraft side window, Atmos. Meas. Tech. Diss., submitted, 2018.