# Peer review of "Remote Sensing of Cloud Droplet Radius Profiles using solar reflectance from cloud sides. Part I: Retrieval development and characterization"

_Atmospheric Measurement Techniques, 2018_

## Referee Comment (RC1) · Anonymous Referee #1 · 19 Sep 2018

**Review of Evald et al. paper "Remote Sensing of Cloud Droplet Radius Profiles using solar reflectance from cloud sides. Part I: Retrieval development and characterization"**

This is a well-written manuscript and I enjoyed reading it. It definitely falls into the scope of AMT and I am sure it will be well referenced by the community. The first cloud side droplet size retrieval papers, I am aware of, are already more than 10 years old and this paper is a very timely. It uses extensively three-dimensional (3D) radiative transfer calculations to study the impact of unknown cloud surface geometry on droplet size retrievals. It is a significant piece of work. I recommend publishing it after minor revision that addresses my questions and suggestion below.

Page 3. Eqs. (1) and (2) are unclear.

Page 5; line 7. I don't think that
Wiscombe,W. and Warren, S.: A Model for the Spectral Albedo of Snow. I: Pure Snow, JAS, 37, 2712–2733,1980
is a good reference for Mie calculations. A much better one is
Wiscombe, W. 1980. "Improved Mie scattering algorithms." *Appl. Opt.*. 1505-1509.

Page 6; line 10. Correct the word 'thie'.

Page 7. Remove the word 'the' in the last paragraph.

Page 10. What are the photon weights?
Page 10. I understood that $<r_{eff}>\_app$ is the apparent effective radius. What is the $<r_{eff}>\_mc$? Is it also apparent radius, as written on page 10?
Page 10; line 13. Fig. 3b shows LWC rather than $r_{eff}$?

Page 12; caption to Fig 7. Figure 7 => Figure 6?
Page 12; Fig 7. What varies here: illumination and viewing angles?

Page 13; Fig 8. What are the gray and color spheres here? Is it the principle plane? What are the green and red dots there?
Page 13; Fig 8. What is the variable here?

Page 16; line 9. I feel like we need a few more sentences here explaining the geometry of gradient classifier *g_class*.
Page 16; line 20. Not clear how D_H=0.25 and D_L=1.5 was found

Page 18; line 4. Did you account for large drops here? In case of large drops $R_{2.1}$ is much smaller and can be confused with shadows. A plot from DISORT with the ratio $R_{0.87}/R_{2.1}$ for $r_{eff}$=12 and 24 um will be helpful here.

Page 19; line 16. When you consider aerosol properties, do they depend on humidity? If not, I'd recommend using aerosols swollen according to the humidity field.

Page 21; Eqs. (13)-(14). Why? Please clarify.

Page 23; Fig. 16 caption. Remove the word 'left'.

Page 23; Fig. 24. Why is the error bar here almost the same for small and large $r_{eff}$? Based on Fig. 16, for large $r_{eff}$ uncertainties are much larger.
Page 25; Fig. 18 caption. You have only panels (a), (b) and (c) here.

---

## Referee Comment (RC2) · Anonymous Referee #2 · 15 Oct 2018

The authors build on prior studies to develop a 3-D effective radius profile retrieval, designed for deployment to field measurements of cloud sides in a follow-up study. The proposed retrieval is comprised of a Bayesian approach and includes means to mitigate the impacts of unknown cloud surface and vertical structure.

The paper is mostly well written and the presented analysis is thorough. In my opinion some of the methods to mitigate 3-D radiative effects and the reasoning for their employment are not sufficiently described and justified, and some minor structural and language issues impair the readability of the work. My recommendation is to accept the manuscript after minor revisions. I include a list of general and specific comments/recommendations that should help the authors in improving upon this first submission.

**General Comments:**

1) The authors included a paragraph in the introduction, which addressed my initial concerns about the manuscript (i.e., novelty). This is well done. However, I feel it would help to summarize the advances in regard to older, similar studies, again in the conclusion. Here, the authors mention that the study advances on prior frameworks, but specifically highlighting the limitations of these older studies and the advances would highlight the impact of the submitted work. One or two sentences should suffice.

2) The paper mentions that the retrieval is designed to drive the retrieval of cloud properties for observations performed with their own specMacs instrument. The authors point out several times that the retrieval conditions are designed to mimic the conditions encountered during these campaigns. Unfortunately, the sampled measurement, field campaign/campaigns and the encountered conditions are never explicitly mentioned anywhere in the paper. From what I can gather from the tables and figures, the observed clouds during the specMacs employment were small cumuli , low flight altitudes of < 2km, cloud top heights between 1.5 and 2km, CCN concentrations up to 1000 cm-3, possible solar zenith angles up to 67deg, the airplane about 2-3 km away from the clouds. Is this correct? From my quick research the measurements were performed with the European HALO aircraft; did you really fly in an altitude of < 2km? I feel it would help to at least mention the campaigns and the overall conditions encountered during the measurement flights.

3) I would recommend to thoroughly proofread the manuscript, if possible by a native English speaker. Some sentences are awkward, there are a number of extra words and punctuation marks that can be removed, some citations are not implemented correctly, and I feel some sentences can either be split up or readability can be improved by commas.

4) Maybe I am missing something, but it is not sufficiently explained and justified, why the gradient classifier g is preferable to the deviation delta_L_2.1. They look similar, but there are obvious differences. Why would g be better than L?

5) The derived moments from the LES droplet size spectra are used to derive the "truth" for the retrieval comparison. Not only would I recommend to avoid scientifically wrong designations like "truth" here, the process of deriving the LES variables for r_eff and LWP is not straightforward and induces more uncertainty; see Alexandrov et al., 2012b; Miller et al., 2016; Zhang et al., 2017, Miller et al., 2018. A lot more information is needed here. How did you average each profile within each height, how did you incorporate weighting functions, etc.

6) It is mentioned several times that the proposed retrieval seems reasonable for optically thick clouds. However, it is never mentioned what this actually means, and whether these conditions are realistic for the expected clouds. At one point, the authors assume tau=500, which from a TOA perspective seems exceptionally high. Is this the regime where we can assume the retrieval to produce reasonable results?

**Specific Comments:**

- Page 1, line 6: remove "with"
- Page 1, line 6: do you mean "small scale cloud heterogeneity"? What do you mean by "small scale structure"?
- Page 1, line 18: Remove the subheading. There is no other subsection in the introduction…
- Page 2, line 35: This sentence is hard to read and a bit awkward. Maybe change to " .., where the observer position is located within the cloud field."
- Page 3, line 6: fix citation
- Page 3, line 7: Which data set do you mean here? This is not about the specMacs measurements, correct? This is the statistical retrieval data set for the Bayesian reff retrieval, correct? Please clarify.
- Page 3, line 12: pixels (plural)
- Page 3, line 15: awkward sentences, maybe add " … and the proposed retrieval is analyzed/tested…"
- Page 4, Eq. 1 and 2: Can you be more precise? These don't really help much…
- Page 5, line 8: fix unit format (1 nm) to be non-italic
- Page 5, line 9: Here, it would help (as mentioned earlier) if we knew more about the specMacs observations, for which the retrieval is designed. You mention that the mid-latitude summer profiles are taken, yet later on you mention that the retrieval should also work for observations in the tropics. Do you adapt the profile for these other measurements?
- Page 5, line 23-24: Is this part of the final retrieval? You perform simulations for two different CCN concentrations, yet this does not seem to be part of the Bayesian approach later on. How do you consider CCN concentrations, if at all?
- Page 6, lines 3-4: fix citation

- Page 6, line 6: So the model includes rain droplets? Maybe mention this explicitly, for some reason I thought this was a typo, because I initially only considered typical effective droplet radii up to 40μm or so…, this raises another question though. How sensitive is the retrieval to rain? Several studies show retrieval issues (TOA, bispectral) when there is precipitation in the clouds, as the vertical profiles and assumptions about gamma-distribution and effective variance fail.
- Page 6, line 26: reff and LWC are already defined
- Page 6, line 29: add comma after "As intended"
- Page 6, line 34: add space between "K" and "km", add comma before "neglecting"
- Figure 2, this figure lacks "a)" and "b)", so the caption is confusing, also the text mentions it shows reff, but it starts with LWP. Overall, very confusing to follow.
- Caption Figure 3: you not only show results for N_CCN = 1000 cm-1, but also 100 cm-1. Also, I am confused by the unit, which should be cm-3, correct? Please clarify.
- Page 8, line 8: remove "the"
- Page 8, line 11: add comma before "which"
- Figure 4: Again, this figures lacks "a)" and "b)", and as far as I can tell, the "b)" part is not discussed at all in the text.
- Page 10, line 4: Change "found" to "retrieved"
- Page 10, Eq 5: change comma to full stop after the equation.
- Page 11, line 4-9: As far as I can tell you don't show any cloud edge reff results. Also, no vertical gradients are shown to evaluate the "better agreement".
- Page 11, lines 14-15. This sentence sounds awkward and I am not sure what you mean here.
- Page 11, line 16: add comma before "where"
- Page 11, line 29: This seems to be a really high optical thickness. Why did you choose that? From a TOA retrieval perspective, this is not really realistic. For comparison: the operational MODIS retrieval stops at tau=150, which is rarely encountered. Does the change in perspective (i.e., cloud side) yield this large limit? Please clarify.
- Page 12, line 9: "optically thick clouds" is very ambiguous. A cirrus is optically thick if tau>5, a stratus with tau> 25, a cumulus with tau>50. Can you be more specific?
- Figure 8: What is shown in the circles? What is the difference between grey and black lines? This Figure is very dense and includes a lot of information, it needs a better description.
- Page 13, last sentence: remove "Figure"
- Page 15, lines 9-10: This sentence is awkward.
- Page 15, line 12: change the last "," to "and" or "to"
- Page 15, line 14: add comma after "Here"
- Page 15, line 16: pixels (plural)
- Page 16, line 11: define CMOS

- Page 16, line 32: pixels (plural)
- Page 17, lines 5-6: I don't understand this sentence.
- Page 18, line 13: add "a" before "viable"
- Page 18, line 14: again, "optically thicker clouds" is ambiguous. Can you be more specific?
- Figure 15: Can you combine Figure 15 and 16, with a panel a) and b)?
- Figure 17: why not show difference maps reff_ret - reff_true? While overall the retrieval seems to work well, differences for individual pixels can be very large.
- Page 25, bottom: 358,000 pixels (plural)
- Figure 18: This is very confusing: Figure 18 is discussed after Figure 20; the discussion includes results shown in Figure 19b. I had trouble following you here.
- Page 27, line 10: profiles (plural)
- Page 27, line 10: What do you mean by true? See also my general comment 5.
- Page 27, line 19: add "the" before "following"
- Page 28, line 3: pixels (plural)
- Page 28, line 4: add "," before "as well"
- Page 28, lines 10-11: remove the description of MYSTIC (already defined)
- Page 29, line 8: Especially in the conclusion you should be more specific about the applicability of the retrieval. What do you mean by "optically thick water clouds"? Are these values realistic, compared to field data? Earlier in the manuscript you mentioned tau=500, which seemed incredibly large to me. Is this the regime, where the retrieval works? When do the uncertainties become too large?
- Page 29, line 12change first "measurements" to "observations"

Literature:

Alexandrov, M. D., Cairns, B., Emde, C., Ackerman, A. S., and van Diedenhoven, B.: Accuracy assessments of cloud droplet size retrievals from polarized reflectance measurements by the research scanning polarimeter, Remote Sens. Environ., 125, 92– 111, https://doi.org/10.1016/j.rse.2012.07.012, 2012b.

Miller, D. J., Zhang, Z., Ackerman, A. S., Platnick, S., and Baum, B. A.: The impact of cloud vertical profile on liquid water path retrieval based on the bispectral method: A theo- retical study based on large-eddy simulations of shallow ma- rine boundary layer clouds, J. Geophys. Res., 121, 4122–4141, https://doi.org/10.1002/2015JD024322, 2016.

Miller, D. J., Zhang, Z., Platnick, S., Ackerman, A. S., Werner, F., Cornet, C., and Knobelspiesse, K.: Comparisons of bispectral and polarimetric retrievals of marine boundary layer cloud microphysics: case studies using a LES–satellite retrieval simulator, Atmos. Meas. Tech., 11, 3689-3715, https://doi.org/10.5194/amt-11-3689-2018, 2018.

Zhang, Z., Dong, X., Xi, B., Song, H., Ma, P. L., Ghan, S. J., Plat- nick, S., and Minnis, P.: Intercomparisons of marine boundary layer cloud properties from the ARM CAP-MBL campaign and two MODIS cloud products, J. Geophys. Res., 122, 2351–2365, https://doi.org/10.1002/2016JD025763, 2017.

---

## Author Comment (AC1) · 31 Dec 2018

**Introduction**

We thank referee #1 for his/her careful reading, comments and suggestions which we address in the following. The authors' answers are printed in italics:

*Remark: The figure and page numbers in the referee comments are corresponding to the original manuscript. If not stated otherwise, figure and equation numbers in the authors' answers are referring to the revised, marked-up manuscript version (showing the changes made) which can be found at the end of this text.*

**General comments**

- This is a well-written manuscript and I enjoyed reading it. It definitely falls into the scope of AMT and I am sure it will be well referenced by the community. The first cloud side droplet size retrieval papers, I am aware of, are already more than 10 years old and this paper is a very timely. It uses extensively three-dimensional (3D) radiative transfer calculations to study the impact of unknown cloud surface geometry on droplet size retrievals. It is a significant piece of work. I recommend publishing it after minor revision that addresses my questions and suggestion below.

  → *Thank you very much for your time and effort in compiling this thorough review! At the end of this text you will find a detailed track change for the revised manuscript.*

- "Page 3. Eqs. (1) and (2) are unclear."

  → *Thank you for this comment. Eqs. (1) and (2) were indeed rather imprecise. Page 3, Eqs. (1) and (2) now read:*

$$p_{\mathsf{fwd}}(L_{0.87}, L_{2.10}, r_{\mathsf{eff}}) = \frac{1}{N}n(L_{0.87}, L_{2.10}, r_{\mathsf{eff}}) \qquad (1)$$

$$p_{\mathsf{fwd}}(r_{\mathsf{eff}}) = \frac{1}{N}n(r_{\mathsf{eff}}). \qquad (2)$$

*Here, the number of radiative transfer results $N$ needs to be large enough for a successful estimation of these two probabilities.*

– "Page 5; line 7. I don't think that

> Wiscombe,W. and Warren, S.: A Model for the Spectral Albedo of Snow. I: Pure Snow, JAS, 37, 2712–2733,1980

is a good reference for Mie calculations. A much better one is

> Wiscombe, W. 1980. "Improved Mie scattering algorithms." Appl. Opt.. "

$\rightarrow$ *Thank you for this hint. We exchanged the quote with you suggested one.*

– "Page 6; line 10. Correct the word 'thie'."

$\rightarrow$ *Done.*

– "Page 7. Remove the word 'the' in the last paragraph."

$\rightarrow$ *Removed.*

– "Page 10. What are the photon weights?"

→ *Thank you for this question. We are using this term in reference to a common methodology in Monte Carlo radiative transfer. In order to speedup Monte Carlo RT computations, each photon carries a photon weight $w_{ph}$ which describes the probability for the current photon path. With each absorption (or scattering) event, this weight is reduced. When photons rech the detector their weight is summed and converted into radiance. This technique is explained in detail in Mayer (2009). We added a corresponding sentence with this citation.*

– "Page 10. I understood that $\langle r_{\text{eff}} \rangle_{app}$ is the apparent effective radius. What is the $\langle r_{\text{eff}} \rangle_{mc}$? Is it also apparent radius, as written on page 10?"

→ *Thanks for this question. Indeed, $\langle r_{\text{eff}} \rangle_{mc}$ is also the apparent radius. Here we just wanted to highlight that the apparent radius is sampled using the photon traces during the Monte Carlo radiative transfer calculations. We now replaced $\langle r_{\text{eff}} \rangle_{mc}$ with $\langle r_{\text{eff}} \rangle_{app}$ to avoid confusion.*

– "Page 10; line 13. Fig. 3b shows LWC rather than reff?"

→ *Thanks for pointing this out. We now refere to Fig. 3a*

– "Page 12; caption to Fig 7. Figure 7 => Figure 6?"

→ *Thank you for spotting that! We changed it accordingly.*

– "Page 12; Fig 7. What varies here: illumination and viewing angles?"

→ *This is correct. Since DISORT is only a 1D model, the cloud surface rotation was simulated by varying illumination and viewing angles. For this*

*reason, we mentioned that we deactivated molecular absorption and scattering to prevent the influence of the longer slanted light paths in the clearsky region above the water cloud. We added following sentence below Fig 7 to clarify this:*

*Page 15, line 15ff:*

> *The radiative transfer calculations for this surface rotation of a water cloud were done with DISORT by varying the illumination and viewing angles while the scattering angle remained fixed.*

– "Page 13; Fig 8. What are the gray and color spheres here? Is it the principle plane? What are the green and red dots there?"
– and –
"Page 13; Fig 8. What is the variable here?"

→ *Thank you for this questions. Since referee #2 (AMTD-2018-234-RC2) also was confused about Fig. 8, we completely restructured this section and disentangled the plots from Fig. 8 (in the old manuscript version) into Fig. 11 and Fig. 12 (in the new manuscript version).*

*The gray spheres are radiance images $L_{0.87}$ of the spherical water cloud, while the colored spheres are images of the radiance ratio $L_{2.10}/L_{0.87}$. We have chosen to show this ratio in Fig. 11 and the 2-wavelength diagram in Fig. 12 to connect the scattered points in Fig. 12 with their spatial location on the sphere in Fig. 11.*

*In addition, we included more color cues and markings to separate the two considered cases ($\vartheta^* = \vartheta_0^*$, direct backscatter, orange color and $\vartheta^* \neq \vartheta_0^*$, not direct backscatter, blue color) in Fig. 7, 9, 11 and 12.*

*Description of Fig. 11 now reads (Page 19, line 7ff):*

*For the direct backscatter geometry ($\vartheta^* = \vartheta_0^*$) on the left and outside the direct backscatter geometry ($\vartheta^* \neq \vartheta_0^*$) on the right, Figure 11a,c show radiance images of $L_{0.87}$ and Figure 11b,d show radiance ratios $L_{2.10}/L_{0.87}$ for the spherical water cloud. The colored radiance ratios will later help to identify regions on the sphere within the 2-wavelength diagram. Furthermore, the two viewing geometries considered in Figure 9 are marked by the green and red dots.*

*For the case with $\vartheta^* \neq \vartheta_0^*$, we furthermore included more markings to separate the two cases ($\vartheta^* > \vartheta_0^*$, oblique viewing angle, red color and $\vartheta^* < \vartheta_0^*$, steep viewing angle, green color) in Fig. 9 and 12.*

*Description of Fig. 12 now reads (Page 19, line 11ff):*

*Figure 12 shows the results in 2-wavelength diagrams for the direct backscatter direction (left) and for a scattering angle of 150° (right). In the 2-wavelength diagrams the radiance pairs from the 3D MYSTIC simulation are shown as scattered points, the results from the one-dimensional DISORT simulations for different effective radii are shown as black lines. Just like in Figures 9 to 11, the large green and red dots in Figure 12 indicate cloud surfaces with same local illumination angle $\vartheta_0^* = 30°$, but steeper ($\vartheta^* < \vartheta_0^*$, green dot) or more oblique local viewing angle ($\vartheta^* > \vartheta_0^*$, red dot).*

*The same (red/green) color scheme is now also used in the following section (4.2.1 Comparison of pixel brightness) to strengthen the connection between the preceding analysis of radiance ambiguities and the mitigation of radiance ambiguities in Section 4.2.1.*

– "Page 16; line 9. I feel like we need a few more sentences here explaining the geometry of gradient classifier $g_{class}$."

→ *Thank you for this suggestion. We completely re-edited this subsection to simplify the explanation of our method:*
*Page 21, line 30ff now read:*

**4.2 Additional information from surrounding pixels**

*(...)*

*Our study will try to find a link between the pixel surrounding and the radiance ambiguity discussed in the preceding section. As discussed in the preceding section, ambiguous radiances are caused by the asymmetric behavior of $L_{0.87}$ and $L_{2.10}$ when changing from a steep local viewing angle towards a more oblique perspective onto a cloud surface. While the pixel brightness $L_{0.87}$ decreases considerably at a more oblique perspective the pixel brightness $L_{2.10}$ increases. While changes in $L_{2.10}$ along a whole cloud profile are associated with a change in effective radius, the geometry-based brightness increase in $L_{2.10}$ should generally occur at smaller scales associated with cloud structures. The method should therefore determine if the pixel is surrounded by darker pixels or surrounded by brighter pixels at $\lambda = 2100\,\mathrm{nm}$. At the same time, the method should be insensitive to instrument noise or Monte Carlo noise between adjacent pixels.*

*Furthermore, we simplified the explanation of the gradient classifier $g_{class}$, which tries to detect this local enhancement of pixel brightness at $\lambda = 2100\,\mathrm{nm}$:*

*Page 22, line 9ff now read:*

**4.2.1. Comparison of pixel brightness**

*To this end, a 2D difference of Gaussians (DoG) filter is used to classify the viewing geometry onto the cloud surface in simulated as well as in measured radiance images. As a 2D difference filter, it compares the brightness of each pixel with the brightness of other pixels in the periphery. The filter consists of two 2D Gaussian functions $G_{\sigma_L}(x,y)$ and $G_{\sigma_H}(x,y)$ with different standard deviations $\sigma_L$ and $\sigma_H$ which specify the inner and outer search radius for the pixel brightness comparison:*

$$G_{\sigma_H}(x,y) = \frac{1}{\sqrt{2\pi\sigma_H^2}} \exp\left(-\frac{x^2 + y^2}{2\sigma_H^2}\right) \tag{11}$$

$$G_{\sigma_L}(x,y) = \frac{1}{\sqrt{2\pi\sigma_L^2}} \exp\left(-\frac{x^2 + y^2}{2\sigma_L^2}\right) \tag{12}$$

*Figure 13a shows the two Gaussians as a function of the angular distance from the considered pixel within the field of view.*

*When the broader kernel $G_{\sigma_L}$ is subtracted from the narrower kernel $G_{\sigma_H}$ (Figure 13a, black line), the average pixel brightness within $\sigma_L$ is compared with the average pixel brightness of the center pixels within $\sigma_H$. This pixel brightness deviation $L_{2.10}^{DoG}$ is obtained by convolving the difference of $(G_{\sigma_H} - G_{\sigma_L})$ with the radiance image $L_{2.10}$:*

$$L_{2.10}^{DoG}(x,y) = (G_{\sigma_H} - G_{\sigma_L}) * L_{2.10}(x,y) \tag{13}$$

*Due to this subtraction, pixels are classified according to their positive or negative radiance deviation compared to their surrounding pixels. By using a not-too-small $\sigma_H$, not only the current pixel but a small surrounding is used, making the method less sensitive to noise of image sensors or Monte Carlo radiative transfer calculations.*

[Figure]

**Figure 13.** *2D Difference of Gaussians $G_{\sigma_H}$ (red) and $G_{\sigma_L}$ (green) which is used as a filter to derive The gradient classifier $g_{class}$ which compares the pixel brightness with the brightness of the surrounding pixels.*

Discussion Paper | Discussion Paper | Discussion Paper | Discussion Paper |

– "Page 16; line 20. Not clear how $D_H = 0.25°$ and $D_L = 1.5°$ was found."

> → *Thank you for pointing that out. We used the radiance deviation $\Delta L_{2.10}$, which we determined in Figure 14a, as a reference for a perfect separation of 3D radiance ambiguities:*
> *Page 24, line 4ff now read:*
>
> > *In the following, this deviation $\Delta L_{2.10}$ is taken as a reference for a perfect separation of 3D radiance ambiguities. It is important to mention, that this deviation $\Delta L_{2.10}$ can only be determined when the effective radius is already known. A method that would yield a similar separation without prior knowledge of $r_{eff}$ could be used as a proxy to mitigate the problem of ambiguous radiances.*
>
> *We then apply the gradient classifier $g_{class}$ to the radiance image $L_{2.10}$. Here, we vary inner and outer search radii systematically. For each set of $\sigma_H$ and $\sigma_L$, we correlate the result from $g_{class}$ with the reference $\Delta L_{2.10}$. The search radii which produce the best correlation between proxy $g_{class}$ and reference $\Delta L_{2.10}$ were $\sigma_H = 0.25°$ and $\sigma_L = 1.5°$:*
> *Page 24, line 8ff now read:*
>
> > *First, an optimal inner and outer search radius $\sigma_H$ and $\sigma_L$ has to be found to use $g_{class}$ as a proxy for the inaccessible radiance deviation $\Delta L_{2.10}$ This optimal search region was found by variation of search radii $\sigma_H$ and $\sigma_L$ and by subsequent correlation of the obtained $g_{class}$ in Figure 15b with the radiance deviation $\Delta L_{2.10}$ in Figure 15a. A maximum correlation with $\Delta L_{2.10}$ was found when the filter operated between $\sigma_H = 0.25°$ and $\sigma_L = 1.5°$.*

– "Page 18; line 4. Did you account for large drops here? In case of large drops $R_{2.1}$ is much smaller and can be confused with shadows. A plot from DISORT with the ratio $R_{0.87}/R_{2.1}$ for $r_{eff} = 12\,\mu m$ and $24\,\mu m$ will be helpful here."

→ *You are absolutely right. We also thought about this problem but did not encounter it during the "Numerical analysis of the retrieval" (Section 6). By manual inspection, we found no major misclassification when we applied the shadow index threshold ($R_{0.87}/R_{2.10} > 3.5$) to cloud side scenes of the clean cloud ensemble with large drops.*

*Nevertheless, your comment made us question our approach. For this reason we followed your suggestion and did some DISORT calculations to analyze our shadow mask. The calculations were done for a water cloud and variable $r_{eff}$ and $\tau_c$. The viewing angle was perpendicular (and thus with high absorption at $2.1\,\mu m$), the illumination angle was $\vartheta_0 = 30°$. The results are presented in Figure 17, where the shadow index is plotted as a function of $r_{eff}$ and $\tau_c$. And, indeed, large drops ($r_{eff} > 12\,\mu m$) can be confused with shadows at higher optical thickness ($\tau_c > 100$). But since most cloud sides of the clean cloud ensemble with $r_{eff} > 12\,\mu m$ had a smaller optical thickness, we did not encounter this misclassification in our 3D Monte Carlo calculations.*

*To prevent this potential misclassification, we now logically combine the shadow index threshold ($R_{0.87}/R_{2.10} > 3.5$) with a simple reflectivity threshold ($R_{2.10} < 0.15$). In this way, only dark **and** highly absorptive cloud regions at $2.10\,nm$ are classified as shadows. Since both masks overlap quite well in Figure 16 and not at all in Figure 17, we found a much better performance with this mask. In our manuscript we changed and added following paragraphs in Section 4.3:*

[Figure]

*Figure 16. (a) Reflectivity at* $2.1\,\mu m$ *for the cloud scene shown in Figure 6 (blue regions mark the simple reflectivity threshold* $R_{2.10} < 0.15$*). (b) Shadow index* $R_{0.87}/R_{2.10}$ *highlighting regions of enhanced cloud absorption caused by multiple diffuse reflections. (red regions mark the shadow index threshold* $R_{0.87}/R_{2.10} > 3.5$*).*

*This enhanced absorption is visible in Figure 16a, where the reflectivity at* $2.1\,\mu m$ *drops considerable for shadowed cloud regions. In Figure 16a, the blue areas illustrate a simple reflectivity threshold* $R_{2.10} < 0.15$*. As a proxy of enhanced absorption, the reflectivity ratio* $R_{0.87}/R_{2.10}$ *(Figure 16b) increases in this regions. In the following, this ratio will be used as shadow index* $R_{0.87}/R_{2.10}$ *to exclude pixels for which light has likely undergone multiple diffuse reflections:*

$$R_{0.87}/R_{2.10} > 3.5 \qquad \text{(shadow index)} \tag{15}$$

*In Figure 16b, the red areas marks regions with* $R_{0.87}/R_{2.10} > 3.5$*. The manual inspection of many cloud scenes confirmed* $3.5$ *as a viable shadow index threshold.*

Discussion Paper | Discussion Paper | Discussion Paper | Discussion Paper |

*Unfortunately, clouds with very large cloud droplets ($r_{\mathrm{eff}} > 12\,\mu m$) can exhibit similar high values of the shadow index. To study this limitation, DISORT calculations were done for an idealized water cloud to characterize the shadow index with respect to cloud optical thickness and effective radius.*

*(... Figure 17 ...)*

*Figure 17 shows the shadow index as a function of effective radius $r_{\mathrm{eff}}$ and optical thickness $\tau_c$ for a geometry ($\vartheta^* = 0°$, $\vartheta_0^* = 30°$) with high absorption at $2.1\,\mu m$. Like in Figure 16, the blue area indicates the simple reflectivity threshold $R_{2.10} < 0.15$ while the red area indicates the shadow index threshold $R_{0.87}/R_{2.10} > 3.5$. Obviously, both shadow thresholds have their disadvantages. At higher optical thickness ($\tau_c > 100$), the shadow index $R_{0.87}/R_{2.10} > 3.5$ can confuse very large cloud droplets ($r_{\mathrm{eff}} > 12\,\mu m$) with cloud shadows. In contrast, the simple reflectivity threshold $R_{2.10} < 0.15$ can misidentify optical thin clouds ($\tau_c < 10$) as cloud shadows. The combined shadow mask $f_{shad}$ of both thresholds in Equation (17) compensates the disadvantage of the shadow index threshold:*

$$f_{shad} = [R_{2.10} < 0.15 \quad and \quad R_{0.87}/R_{2.10} > 3.5] \quad \text{(shadow mask)}$$
(16)

*In this way, only dark ad highly absorptive cloud regions at $2.10\,nm$ are classified as shadows.*

Discussion Paper | Discussion Paper | Discussion Paper | Discussion Paper |

[Figure]

*Figure 17.* Shadow index $R_{0.87}/R_{2.10}$ for water clouds as a function of effective radius $r_{\text{eff}}$ and cloud optical thickness $\tau_c$ for a geometry $(\vartheta^* = 0°, \vartheta_0^* = 30°)$ with high absorption at $2.1\,\mu m$. Like in Figure 16, the blue area indicates the simple reflectivity threshold $R_{2.10} < 0.15$ while the red area indicates the shadow index threshold $R_{0.87}/R_{2.10} > 3.5$.

– "Page 19; line 16. When you consider aerosol properties, do they depend on humidity? If not, I'd recommend using aerosols swollen according to the humidity field."

$\rightarrow$ *Thanks for this question. As written in the manuscript*
*On Page 28, line 5:*

*Atmospheric aerosol was included by using the continental average mixture from the Optical Properties of Aerosols and Clouds (OPAC) package (Hess et al., 1998).*

*More than half of all particles within the used continental average mixture is able to take up water.*

*As (Hess et al., 1998) states:*

> *(...) For those aerosols that are able to take up water, the mode radius as well as the limiting radii are increased with increasing relative humidity (...)*

*The RT code MYSTIC is using the humidity field to apply this modification.*

– "Page 21; Eqs. (13)-(14). Why? Please clarify."

  → *Yes, you are right - in the original manuscript Eqs. (13)-(14) were not really well explained. For this reason, we re-arranged this equation to simplify the explanation:*

  *Page 30, line 5ff now read:*

  > *The flipped cloud microphysics were derived by taking the additive inverse $-r_{\text{eff}}^{orig}$ of the original effective radius fields and add an offset $r_{\text{eff}}^{offset}$:*
  >
  > $$r_{\text{eff}}^{flip} = -r_{\text{eff}}^{orig} + r_{\text{eff}}^{offset}. \tag{17}$$
  >
  > *To ensure positive and realistic values for $r_{\text{eff}}^{flip}$, the offset $r_{\text{eff}}^{offset}$ was chosen to be at least $4\,\mu$m larger than the largest values found in all cloud fields. Thus, $r_{\text{eff}}^{offset} = 12\,\mu\text{m} + 4\,\mu\text{m} = 16\,\mu\text{m}$ was used in Equation (18) for the polluted cloud ensemble ($CCN = 1000\,\text{cm}^{-3}$) and $r_{\text{eff}}^{offset} = 22\,\mu\text{m} + 4\,\mu\text{m} = 26\,\mu\text{m}$ for the clean cloud ensemble ($CCN = 100\,\text{cm}^{-3}$).*

Discussion Paper | Discussion Paper | Discussion Paper | Discussion Paper |

> *To preserve the optical thickness $\tau^{orig}$ of the original cloud field,*
>
> $$\tau^{flip} \equiv \tau^{orig}, \tag{18}$$
>
> *the well established relationship in Equation (20) was used to derive the liquid water content $LWC^{flip}$ for the flipped cases in Equation (21):*
>
> $$\tau \propto \frac{LWC}{r_{eff}}, \tag{19}$$
>
> $$LWC^{flip} = \frac{r_{eff}^{flip}}{r_{eff}^{orig}} LWC^{orig}. \tag{20}$$

– "Page 23; Fig. 16 caption. Remove the word 'left'."

> → *Done.*

– "Page 25; Fig. 17. Why is the error bar here almost the same for small and large reff? Based on Fig. 16, for large reff uncertainties are much larger."

> → *This comment proofed to be very helpful! Your question made us question our implementation of Eq. 19. (now Eq. 22) to calculate the standard deviation. To our surprise, we found a missing bracket and corrected our code. We recalculated and updated our results for Fig. 17 and Fig. 18. For a clearer presentation, the mean statistical retrieval uncertainty $\sigma(r_{eff})$ derived by Eq. 19. (now Eq. 22) is now shown by red error bars in Fig. 17 and Fig. 18.*
> *Added to description of Fig 16. (now Fig 19b), page 32, line 10:*

> *Interestingly, $\sigma(r_{\text{eff}})$ increases with the effective radius from $\sigma(r_{\text{eff}} = 6\,\mu m) = \pm 1\,\mu m$ to $\sigma(r_{\text{eff}} = 16\,\mu m) = \pm 3\,\mu m$.*

*Added to description of Fig 17. (now Fig 20), page 33, line 18ff:*

> *Like in Figure 19b, $\sigma(r_{\text{eff}})$ increases with $r_{\text{eff}}$ from $\sigma(r_{\text{eff}} = 6\,\mu m) = \pm 1\,\mu m$ to $\sigma(r_{\text{eff}} = 12\,\mu m) = \pm 2\,\mu m$. Within $\sigma(r_{\text{eff}})$, the retrieval reproduces the mean effective radius profile for all three cases quite well. For some specific cloud regions, however, there are also large differences of up to $\pm 3\,\mu m$. This is especially true at cloud edges and close to shadows.*

*Added to description of Fig 18. (now Fig 23), page 36, line 4ff:*

> *For this scene with overall larger $r_{\text{eff}}$, the retrieval underestimates $r_{\text{eff}}$ by $\pm 2.5\,\mu m$ at the upper cloud side part. Nevertheless, the statistical retrieval detects the mean effective radius profiles well within $\sigma(r_{\text{eff}})$. Like in Figure 20a, $\sigma(r_{\text{eff}})$ increases with $r_{\text{eff}}$ from $\sigma(r_{\text{eff}} = 12\,\mu m) = \pm 2\,\mu m$ to $\sigma(r_{\text{eff}} = 21\,\mu m) = \pm 3\,\mu m$.*

– "Page 25; Fig. 18 caption. You have only panels (a), (b) and (c) here."

→ *Thank you for your sharp eye! We changed Fig. 18 caption accordingly.*

**References**

[revised manuscript text omitted]

$$\sout{2}p_{\text{fwd}}(L_{0.87}, L_{2.10}, r_{\text{eff}}) \quad \propto n(L_{0.87}, L_{2.10}, r_{\text{eff}})$$

$$p_{\text{fwd}}(r_{\text{eff}}) \quad \propto n(r_{\text{eff}}) = \sum_{L_{0.87}} \sum_{L_{2.10}} n.$$

$$p_{\text{fwd}}(L_{0.87}, L_{2.10}, r_{\text{eff}}) = \frac{1}{N} n(L_{0.87}, L_{2.10}, r_{\text{eff}}) \tag{1}$$

$$p_{\text{fwd}}(r_{\text{eff}}) = \frac{1}{N} n(r_{\text{eff}}). \tag{2}$$

Here, the number of radiative transfer results  $N$ needs to be large enough for a successful estimation of these two probabilities. Simultaneously, the forward simulation has to cover all values expected in the real world application. With the likelihood probability $p(L_{0.87}, L_{2.10} | r_{\text{eff}})$ as a conditional probability, it can be written as the quotient of the joined probability $p_{\text{fwd}}(L_{0.87}, L_{2.10}, r_{\text{eff}})$ and $p_{\text{fwd}}(r_{\text{eff}})$  from Equation (1) and Equation (2):

$$p(L_{0.87}, L_{2.10} | r_{\text{eff}}) = \frac{p_{\text{fwd}}(L_{0.87}, L_{2.10}, r_{\text{eff}})}{p_{\text{fwd}}(r_{\text{eff}})} \tag{3}$$

$$p(r_{\text{eff}} | L_{0.87}, L_{2.10}) = \frac{p(L_{0.87}, L_{2.10} | r_{\text{eff}}) \, p_{\text{pr}}(r_{\text{eff}})}{\int p(L_{0.87}, L_{2.10} | r_{\text{eff}}) \, p_{\text{pr}}(r_{\text{eff}}) \, dr_{\text{eff}}} \tag{4}$$

[revised manuscript text omitted]

$$2 r_{\text{eff}}^{\text{flip}} = 16\,\mu\text{m} - r_{\text{eff}}^{\text{orig}} \qquad (\text{CCN} = 1000\,\text{cm}^{-1})$$

$$r_{\text{eff}}^{\text{flip}} = 26\,\mu\text{m} - r_{\text{eff}}^{\text{orig}} \qquad (\text{CCN} = 100\,\text{cm}^{-1})$$

fields and add an offset $r_{\text{eff}}^{\text{offset}}$:

$$r_{\text{eff}}^{\text{flip}} = -r_{\text{eff}}^{\text{orig}} + r_{\text{eff}}^{\text{offset}}. \tag{17}$$

 To ensure positive and realistic values for $r_{\text{eff}}^{\text{flip}}$, the offset $r_{\text{eff}}^{\text{offset}}$ was chosen to be at least 4 µm larger than the  largest values found in  all cloud fields. Thus, $r_{\text{eff}}^{\text{offset}} = 12\,\mu\text{m} + 4\,\mu\text{m} = 16\,\mu\text{m}$ was used in Equation (17) for the polluted cloud ensemble (CCN = 1000 cm$^{-3}$) and $r_{\text{eff}}^{\text{offset}} = 22\,\mu\text{m} + 4\,\mu\text{m} = 26\,\mu\text{m}$ for the clean cloud ensemble (CCN = 100 
[revised manuscript text omitted]

---

## Author Comment (AC2) · 31 Dec 2018

**Introduction**

We thank referee #2 for his/her careful reading, comments and suggestions which we address in the following. The authors' answers are printed in italics:

*Remark: The figure and page numbers in the referee comments are corresponding to the original manuscript. If not stated otherwise, figure and equation numbers in the authors' answers are referring to the revised, marked-up manuscript version (showing the changes made) which can be found at the end of this text.*

**General comments**

- **–** "The authors included a paragraph in the introduction, which addressed my initial concerns about the manuscript (i.e., novelty). This is well done. However, I feel it would help to summarize the advances in regard to older, similar studies, again in the conclusion. Here, the authors mention that the study advances on prior frameworks, but specifically highlighting the limitations of these older studies and the advances would highlight the impact of the submitted work. One or two sentences should suffice."

  → *Thank you very much for your time and effort in compiling this thorough and detailed review! At the end of this text you will find a detailed track change for the revised manuscript. We agree that such a conclusion was missing so far in our manuscript. For this reason, we added and modified the following paragraph in the conclusion section:*
  *Page 37, line 8ff:*

*Up to now, their approach could not directly applied to realistic cloud side measurements (e.g. specMACS on HALO) since their studies lack the varying geometries of an airborne perspective. Furthermore, the effective radius was only parameterized and not directly calculated by a microphysical model. Moreover, Zinner et al. (2008) used the line of sight method to associate the forward modeled radiance with $r_{\text{eff}}$ found at the first cloudy grid box. To advance the technique to realistic airborne measurements, this study addressed following scientific objectives to overcome these limitations:*

1. *Extend the existing approach to realistic airborne perspectives and develop methods to test the sensitivity of reflected radiances from cloud sides to cloud droplet radius, where the observer position is located within the cloud field. To this end, methods were developed to identify suitable observation positions within a model cloud field and to calculate an apparent effective radius for each forward modeled sensor pixel.*

2. *In this course, 3D radiative effects caused by the unknown cloud surface orientation were investigated. A technique was proposed to mitigate their impact on cloud droplet size retrievals by putting pixel in context with their surrounding.*

3. *Finally, an effective radius retrieval for the cloud side perspective was developed and tested for cloud scenes which were used during the retrieval development, as well as for unknown scenes.*

*(...)*

Discussion Paper | Discussion Paper | Discussion Paper | Discussion Paper |

– "The paper mentions that the retrieval is designed to drive the retrieval of cloud properties for observations performed with their own specMacs instrument. The authors point out several times that the retrieval conditions are designed to mimic the conditions encountered during these campaigns. Unfortunately, the sampled measurement, field campaign/campaigns and the encountered conditions are never explicitly mentioned anywhere in the paper. From what I can gather from the tables and figures, the observed clouds during the specMacs employment were small cumuli, low flight altitudes of $< 2\,\mathrm{km}$, cloud top heights between $1.5$ and $2\,\mathrm{km}$, CCN concentrations up to $1000\,\mathrm{cm}^{-3}$, possible solar zenith angles up to $67\,°$, the airplane about $2 - 3\,\mathrm{km}$ away from the clouds. Is this correct? From my quick research the measurements were performed with the European HALO aircraft; did you really fly in an altitude of $< 2\,\mathrm{km}$? I feel it would help to at least mention the campaigns and the overall conditions encountered during the measurement flights."

→ *Thank you for this comments and questions. You are right in your assumption that the proposed retrieval is focused on the liquid part of convective liquid water clouds, e.g., Cumuli and Cumuli congesti and was designed for specMACS on HALO. As you have already guessed, specMACS measurements of these clouds were performed during the ACRIDICON-CHUVA campaign (Wendisch et al., 2016). Regarding the flight altitude of around* $2\,\mathrm{km}$*, we refer you to Fig. 7 in Wendisch et al. (2016). Before the developing convection was probed in profile flights, the aerosol background and the small-scale convection was probed in their early stages in low-level flight legs between* $1\,\mathrm{km}$ *to* $3\,\mathrm{km}$ *altitude. To introduce the encountered conditions, we now added following paragraph to Section 1.1 ("Scientific objectives and scope of this work"):*

*Page 3, line 19ff:*

*The target of this work is the liquid part of convective liquid water clouds, e.g., Cumulus mediocris, Cumulus congestus and Trade-wind cumulus, which exhibit well-developed cloud sides. During September 2014, images of such cloud sides were acquired with the **spec***trometer of the* **M***unich* **A***erosol* **C***loud* **S***canner (spec-MACS, Ewald et al., 2016) over the Amazonian rainforest near Manaus, Brazil. The measurements were performed during the ACRIDICON-CHUVA campaign (Wendisch et al., 2016) during which the specMACS instrument was deployed on the German research aircraft HALO (Krautstrunk and Giez, 2012), mounted in a side-looking configuration. The campaign focused on aerosol-cloud-precipitation interactions over the Amazon rain forest. More specific, the campaign investigated the impact of wildfire aerosols on cumulus clouds and on their later development into deep convection. During the campaign flights, the aerosol background and the small-scale convection in their early stages was probed in low-level flight legs between $1\,\text{km}$ to $3\,\text{km}$ altitude. At cloud base level, mean cloud condensation nuclei (CCN) concentrations ranged between $250\,\text{cm}^{-3}$ and $2000\,\text{cm}^{-3}$ (Andreae et al., 2018). The spec-MACS measurements were done of smaller cumulus clouds in a distance of $2\,\text{km}$ to $6\,\text{km}$ and with top heights between $1.5\,\text{km}$ and $3\,\text{km}$. Subsequently, vertical profile flights were performed to measure the microphysical properties of the developing convection in-situ. This manuscript (Part 1) develops a statistical effective radius retrieval for these non-glaciated cumulus clouds which where measured during the low-level flights. (...)*

Discussion Paper | Discussion Paper | Discussion Paper | Discussion Paper |

> *We hope that this paragraph now gives the reader a better understanding of the conditions the retrieval was designed for.*

- "I would recommend to thoroughly proofread the manuscript, if possible by a native English speaker. Some sentences are awkward, there are a number of extra words and punctuation marks that can be removed, some citations are not implemented correctly, and I feel some sentences can either be split up or readability can be improved by commas."

  → *Thank you for this suggestion. We went through the manuscript once again to correct for grammar, spelling, wrong citations and punctuation. To further improve the readability, a native English speaker re-edited the manuscript. For a detailed track change, the reader is referred to the revised, marked-up manuscript version at the end of this text.*

- "The derived moments from the LES droplet size spectra are used to derive the "truth" for the retrieval comparison. Not only would I recommend to avoid scientifically wrong designations like "truth" here, the process of deriving the LES variables for $r_{eff}$ and LWP is not straightforward and induces more uncertainty; see Alexandrov et al. (2012); Miller et al. (2016); Zhang et al. (2017), Miller et al. (2018). A lot more information is needed here. How did you average each profile within each height, how did you incorporate weighting functions, etc."

  → *Thank you very much for pointing this out. We are fully aware of the complicated process of deriving LES variables for $r_{eff}$. We are confident that we correctly addressed all required steps to infer $r_{eff}$, e.g. the step to convert LES moments to $r_{eff}$ and the step to determine the apparent effective radius of cloud sides in case of inhomogeneous cloud microphysics. Your comment made us realize that we should improve the description of these*

*two steps to enhance the reproducibility of our study. For this reason, we now included Equations 5-9 in Section 2.4 ("Cumulus cloud model") to describe the conversion from the model particle size distribution to model effective radius $r_{\text{eff}}$ for each model grid box:*

*Page 7, line 21ff:*

> *Using Equation 5-8, effective radius $r_{\text{eff}}$, liquid water content LWC and total cloud droplet concentration $N_d$ can be calculated from mass mixing ratios $m_i$ in $\text{g kg}^{-1}$ and cloud droplet mixing ratios $n_i$ in $\text{kg}^{-1}$ given for the 33 LES size bins:*
>
> $$r_i(x,y,z) = \sqrt[3]{\frac{m_i(x,y,z)}{n_i(x,y,z)} \frac{3}{4\pi\rho_w}}, \tag{1}$$
>
> $$r_{\text{eff}}(x,y,z) = \frac{\sum_{i=1}^{33} r_i^3(x,y,z)\, n_i(x,y,z)\, \Delta r_i}{\sum_{i=1}^{33} r_i^2(x,y,z)\, n_i(x,y,z)\, \Delta r_i}, \tag{2}$$
>
> $$LWC(x,y,z) = \sum_{i=1}^{33} m_i(x,y,z)\rho_{\text{air}}(x,y,z), \tag{3}$$
>
> $$N_d(x,y,z) = \sum_{i=1}^{33} n_i(x,y,z)\rho_{\text{air}}(x,y,z). \tag{4}$$

*To improve upon the description of the second step (the conversion from model effective radius $r_{\text{eff}}$ to visible $\langle r_{\text{eff}}\rangle_{app}$ at cloud side), we now try to better introduce our approach in Section 3.2 ("Determination of the apparent effective radius") to obtain $\langle r_{\text{eff}}\rangle_{app}$ during the Monte Carlo tracing of photons:*

Discussion Paper | Discussion Paper | Discussion Paper | Discussion Paper | Discussion Paper

**3.2 Determination of the apparent effective radius**

*As various studies have pointed out, the process of deriving the LES variables for $r_{\text{eff}}$ in the first place is not straightforward (Alexandrov et al., 2012; Miller et al., 2016; Zhang et al., 2017; Miller et al., 2018). First, $r_{\text{eff}}$ has to be derived from model parameters which describe the particle size distribution. This step was explained in Section 2.4 by Equation 5-8. Secondly, an approach to infer the visible effective radius has to be developed in case of inhomogeneous cloud microphysics. In their statistical retrieval approach, Zinner et al. (2008) traced along the line of sight of each sensor pixel until hitting the first cloudy model grid box from which they selected their $r_{\text{eff}}$ corresponding to the observed radiances. This method has its limitations when it comes to highly structured cloud sides with horizontally inhomogeneous microphysics.*

*In our approach, we calculate radiances $L_{0.87}$, $L_{2.10}$ and the apparent effective radius $\langle r_{\text{eff}} \rangle_{app}$ for each pixel. For this reason, we do not have to average each profile within each height. In contrast, we have a direct co-registration of radiances and microphysics. In addition, a weighting function is directly incorporated by the Monte Carlo tracing of photons to obtain the apparent effective radius"*
*This is explained on page 12, line 17ff:*

Discussion Paper | Discussion Paper | Discussion Paper | Discussion Paper | Discussion Paper |

> *The co-registration of responsible cloud droplet sizes with modeled radiances is essential. Besides the observation perspective, this apparent effective radius $\langle r_{\text{eff}} \rangle_{app}$ also depends on the observed wavelength since different scattering and absorption coefficients lead to different cloud penetration depths. In the following, a technique will be introduced to obtain $\langle r_{\text{eff}} \rangle_{app}$ during the Monte Carlo tracing of photons. As discussed by Platnick (2000), there exist analytical as well as statistical methods to consider the contribution of each cloud layer to the apparent effective radius $\langle r_{\text{eff}} \rangle_{app}$. Advancing the one-dimensional weighting procedures of Platnick (2000) and Yang et al. (2003), the 3D tracing of photons in MYSTIC is utilized to calculate the optical properties of inhomogeneous, mixed-phase clouds. The apparent effective radius $\langle r_{\text{eff}} \rangle_{ph}$ for a photon is a weighted, linear combination of the individual effective radii $r_{\text{eff}}$ the photon encounters on its path through the cloud: (...)*

*Finally, we replaced the misleading term "true effective radius" with "apparent effective radius" throughout the manuscript.*

– "It is mentioned several times that the proposed retrieval seems reasonable for optically thick clouds. However, it is never mentioned what this actually means, and whether these conditions are realistic for the expected clouds. At one point, the authors assume tau=500, which from a TOA perspective seems exceptionally high. Is this the regime where we can assume the retrieval to produce reasonable results?"

$\rightarrow$ *Thank you for this observation. By answering your question we realized that we have to be more specific regarding the term "optically thicker" and "optically thick". In addition, we forgot to summarize the retrieval*

*specifications in our conclusions.*

*As a thought experiment, we use a "optically thick" water cloud with $\tau_c = 500$) in Section 4.2, to dissect and study the impact of an unknown cloud surface orientation. The retrieval, however, is designed to produce reasonable results for cumuli with $\tau_c = 15 - 150$ and well-developed cloud sides, e.g. Cumuli mediocris, Cumuli congestus and Trade-wind cumuli. Here, the optical thickness range is defined by the LES cloud model ensemble. In Figure 19, we bring together both studies and compare the ambiguity of radiance from the realistic LES cloud scenes (2D histogram plot) with the ambiguity of radiance from the spherical cloud with $\tau_c = 500$. Here, we note that ...*

*Page 31, Line 4ff:*

> *... the radiance spread from the three-dimensional model cloud sides, for the most part, can be explained by the one-dimensional DISORT results for $\tau_c = 500$ (dashed line for variable cloud surface inclination).*

*We now reason and clarify our definition of "optically thicker" and "optically thick" for these two cases:*

*Page 14, Line 16ff:*

Discussion Paper | Discussion Paper | Discussion Paper | Discussion Paper |

> *By "optically thicker", we refer to cumuli contained in the LES model output which exhibit well-developed cloud sides, e.g. like Cumuli mediocris, Cumuli congestus and Trade-wind cumuli. To give a concrete example, this term includes clouds with $\tau_c > 15$, e.g. with an average LWC of $0.5\,\mathrm{g\,m^{-3}}$, $r_{eff} = 10\,\mu m$ and with a vertical extent of $200\,m$ and onward. Since the maximum optical thickness contained in the LES output is $\tau_c = 176$, the retrieval is designed for cumuli with $\tau_c = 15 - 150$. To dissect the impact of an unknown cloud surface orientation on the effective radius retrieval, the following study will use a "optically thick" water cloud ($\tau_c = 500$). We subsequently develop a method to exclude cloud shadows and to mitigate radiance ambiguities for the cumulus clouds contained in the LES ensemble using the obtained insights.*

*Furthermore, the conclusion now summarizes the regime for which the retrieval was designed:*

*Page 38, Line 20ff:*

> *Defined by the used LES model fields and the chosen geometries of the forward simulations, this retrieval is designed for*
>   - *cloud side measurements of the liquid part of convective water clouds, e.g., Cumulus mediocris, Cumulus congestus and Trade-wind cumulus, which exhibit well-developed cloud sides.*
>   - *cloud tops between $1.5 - 2$ km, an optical thickness between $15 - 150$ and effective radii between $4 - 24$ µm,*
>   - *spatially highly resolved ($10 \times 10$ m) images of the spectral radiance at $\lambda = 870$ nm and $\lambda = 2100$ nm,*
>   - *a variable and unknown CCN background concentration between $100 - 1000$ cm$^{-3}$,*
>   - *variable sun zenith angles $\vartheta_0$ between $7 - 67°$ for tropical as well as mid-latitude application,*
>   - *an airborne perspective at $1.7$ km altitude with a field of view of $\Delta\varphi = 46°$ (azimuthal) and $\Delta\vartheta = 40°$ (horizontal) centered $5°$ below the horizon.*

**Specific comments**

  – "Page 1, line 6: remove "with""

  → *Thank you for this find. Removed.*

  – "Page 1, line 6: do you mean "small scale cloud heterogeneity"? What do you mean by "small scale structure"?"

> → *This is correct. We changed this formulation from "... faced with the small scale structure of cloud sides ..." to "... faced with the small-scale heterogeneity of cloud sides ...".*

– "Page 1, line 18: Remove the subheading. There is no other subsection in the introduction..."

> → *Thank you for this comment. We now renamed the "Introduction" into "1. Current state of passive remote sensing of clouds" and included the unnumbered subsection "Scientific objectives and scope of this work". By that way, we separate the paragraph containing the literature research from the paragraph introducing the scientific objectives and the novelty of our work.*

– "Page 2, line 35: This sentence is hard to read and a bit awkward. Maybe change to " .., where the observer position is located within the cloud field.""

> → *Thanks! The sentence now reads:*
>
> *Page 3, line 13ff:*
>
> > *Extend the existing approach to realistic airborne perspectives and develop methods to test the sensitivity of reflected radiances from cloud sides to cloud droplet radius, where the observer position is located within the cloud field.*

– "Page 3, line 6: fix citation"

> → *Fixed citation.*

**–** "Page 3, line 7: Which data set do you mean here? This is not about the specMacs measurements, correct? This is the statistical retrieval data set for the Bayesian reff retrieval, correct? Please clarify."

→ *This is correct. In Section 2, we introduce the new LES cloud model data set which is used for the Monte Carlo forward calculations. For this reason, the sentence now reads:*

*Page 4, line 1:*

*Section 2 shortly recapitulates established methods and introduces the new cloud model data set with explicit cloud microphysics. (...)*

**–** "Page 3, line 12: pixels (plural)"

→ *Thank you for this comment. We corrected this mistake at several locations throughout the manuscript.*

**–** "Page 3, line 15: awkward sentences, maybe add " ... and the proposed retrieval is analyzed/tested...""

→ *Thank you. We split the sentence into two, which now read:*

*Page 4, line 9ff:*

*Finally, the developed retrieval is tested in Section 6 with unknown scenes of cloud sides and different aerosol backgrounds. Furthermore, the retrieval is analyzed for potential biases.*

– "Page 4, Eq. 1 and 2: Can you be more precise? These don't really help much..."

→ *You are right. This was also pointed out by referee #1 (AMTD-2018-234-RC1). Eqs. (1) and (2) were indeed rather imprecise.*

*Page 3, Eqs. (1) and (2) now reads:*

$$p_{\text{fwd}}(L_{0.87}, L_{2.10}, r_{\text{eff}}) = \frac{1}{N} n(L_{0.87}, L_{2.10}, r_{\text{eff}}) \tag{5}$$

$$p_{\text{fwd}}(r_{\text{eff}}) = \frac{1}{N} n(r_{\text{eff}}). \tag{6}$$

*Here, the number of radiative transfer results $N$ needs to be large enough for a successful estimation of these two probabilities.*

– "Page 5, line 8: fix unit format (1 nm) to be non-italic"

→ *Corrected.*

– "Page 5, line 9: Here, it would help (as mentioned earlier) if we knew more about the specMacs observations, for which the retrieval is designed. You mention that the mid-latitude summer profiles are taken, yet later on you mention that the retrieval should also work for observations in the tropics. Do you adapt the profile for these other measurements?"

→ *Thank you for this question. As suggested earlier we now describe the encountered campaign conditions in the introduction section. For the LES simulations (Jiang and Li, 2009), a mean thermodynamic sounding from the Rain In Cumulus over the Ocean (RICO) field experiment (Rauber et al., 2007) was used. Regarding air temperature and water*

*vapor concentration, this profile is in between the tropical and the mid-latitude summer profiles by Anderson et al. (1986). Since gaseous absorption is negligible at the chosen wavelength region of $(870.0 \pm 0.6)$ nm and $(2100.0 \pm 3.3)$ nm and since the pressure profiles are almost identical we have chosen the mid-latitude summer profile. This choice still allows for a tropical as well as a mid-latitude application of the retrieval without the need to adapt the atmospheric profile. We added following sentence to our manuscript:*

*Page 6, line 11ff:*

> *Since gaseous absorption is negligible at the chosen wavelength region of $(870.0 \pm 0.6)$ nm and $(2100.0 \pm 3.3)$ nm, this choice still allows for a tropical as well as a mid-latitude application of the retrieval.*

– "Page 5, line 23-24: Is this part of the final retrieval? You perform simulations for two different CCN concentrations, yet this does not seem to be part of the Bayesian approach later on. How do you consider CCN concentrations, if at all?"

  $\rightarrow$ *Thank you for this question. This made use realize that the combination of different CCN concentrations into the same lookup table was only implicitly described and needs a better introduction! It is important to emphasize that the retrieval is designed to be independent from a-priori knowledge of $N_{\mathrm{CCN}}$. Otherwise, retrieved $r_{\mathrm{eff}}$ could be biased by measured CCN concentrations which would made the retrieval unsuitable for aerosol-cloud-interactions studies. For this reason, we now mention this earlier in our manuscript.*

*Page 4, line 7ff now reads:*

> *In contrast to previous studies, different aerosol backgrounds are now also considered. For the retrieval, the results for different CCN concentrations are combined within one lookup table to be independent from a-priori knowledge of $N_{\text{CCN}}$. (...)*

*Where we describe the "Construction of the lookup table" in Section 5.2, we now motivate and explicitly describe this combination of results within the same histogram:*

*Page 29, line 1ff:*

> **5.2 Construction of the lookup table**
>
> *In the next step, simulated radiances were binned into a multidimensional histogram with equidistant steps in $L_{0.87}$, $L_{2.10}$, $r_{\text{eff}}$, $\vartheta$ and $g_{\text{class}}$. Here, it is important to emphasize that the retrieval is designed to be independent from a-priori knowledge of $N_{\text{CCN}}$. If the posterior distributions would separate between clean and polluted cases, the retrieval could tend to larger $r_{\text{eff}}$ when a low $N_{\text{CCN}}$ is measured. Such an retrieval would be unsuitable to study aerosol-cloud-interactions. For this reason, the radiance results from the polluted and the clean cloud ensemble are combined within the same histogram.*

– "Page 6, lines 3-4: fix citation"

→ *Fixed citation.*

– "Page 6, line 6: So the model includes rain droplets? Maybe mention this explicitly, for some reason I thought this was a typo, because I initially only considered typical effective droplet radii up to 40 μm or so..., this raises another question though. How sensitive is the retrieval to rain? Several studies show retrieval issues (TOA, bispectral) when there is precipitation in the clouds, as the vertical profiles and assumptions about gamma-distribution and effective variance fail."

→ *Thank you very much for addressing this issue! As authors of Zinner et al. (2010), we investigated the impact of drizzle on effective radius retrievals and came to following conclusion:*

*Zinner et al. (2010) on page 13, right column, first paragraph:*

> *(...) We investigate two different types of boundary layer clouds: (1) a case of a drizzling fully overcast marine stratocumulus deck at two stages during a diurnal cycle; and (2) a more complex cloud scene of a drizzling cumulus field. For both cloud types the impact of drizzle formation on the MODIS retrieval is very small. The sensitivity to the drizzle size drops in the cloud deck is too small to explain contrasts like that seen in Fig. 1 let alone a clear detection of drizzle. (...)*

*Further studies (Nakajima et al., 2009; Zhang et al., 2012) also found only a small impact of drizzle on bi-spectral methods of about $0.5 - 2\,\mu m$. Zhang et al. (2012), for example, found that*

*On page 10, left column:*

> *(...) drizzle drops with $r_{eff} > 30\,\mu m$ have a very minor impact on the $r_{e2.1}$ and $\tau$ retrievals (...)*

*These studies were done for smaller cumulus clouds with light or moderate rain rates of $0.05 - 1\,mm\,h^{-1}$. As our study also focused on shallow*

*cumulus convection, we went along with these findings and did no further sensitivity study regarding rain. For higher precipitation rates, however, the more recent study by Zhang (2013) suggested more detailed studies to assess the impact of rain on bi-spectral retrieval techniques. For this reason, we added following paragraph in our conclusion to state this limitation and to suggest further studies:*

*Page 39, line 12ff now read:*

> *This study did not address the open question if and how strong rain can influence the retrieval performance. Several studies (Nakajima et al., 2009; Zinner et al., 2010; Zhang et al., 2012) found only a small impact of drizzle on bi-spectral methods of about $0.5 - 2\,\mu m$. Although beyond the scope of this work, subsequent studies should address the influence of rain on bi-spectral retrievals for stronger precipitation rates as suggested by the more recent study of Zhang (2013).*

– "Page 6, line 26: reff and LWC are already defined"

   → *That is correct. We removed this redundant definition here.*

– "Page 6, line 29: add comma after "As intended""

   → *Thanks, done.*

– "Page 6, line 34: add space between "K" and "km", add comma before "neglecting""

   → *Done.*

**–** "Figure 2, this figure lacks "a)" and "b)", so the caption is confusing, also the text mentions it shows reff, but it starts with LWP. Overall, very confusing to follow."

> → *Thank you for this feedback. We now divided this figure in panel group (A) and (B) and labeled each panel accordingly:*
>
> *Caption of Figure 2 now reads:*
>
> > *(A) Snapshot of LES cloud fields at* 12 h 40 min *LT with (A1) liquid water path in* $g\,m^{-2}$ *and (A2) north-south and (A3) east-west cross-sections of the liquid water content field in* $g\,m^{-3}$. *(B) Same LES snaphop with (B1) optical thickness* $\tau$ *and (B2) north-south and (B3) east-west cross-sections of effective radius* $r_{\text{eff}}$ *in* $\mu$. *The insets in A3 and B3 contain zoomed cross-sections of LWC and* $r_{\text{eff}}$ *for a cloud edge region showing signs of lateral entrainment.*
>
> *We also revised the description of Figure 2 in the text of our manuscript.*

**–** "Caption Figure 3: you not only show results for $N_{\text{CCN}} = 1000\,\text{cm}^{-1}$, but also $100\,\text{cm}^{-1}$. Also, I am confused by the unit, which should be cm$^{-3}$, correct? Please clarify."

> → *Thank you for pointing this out. The unit was indeed a typo which we fixed throughout the manuscript. We explain Figure 3 now more detailed:*
>
> *Caption of Figure 3 now reads:*

> *The contoured frequency by altitude diagrams (CFADs) show (a) effective radius $r_{\text{eff}}$, (b) liquid water content LWC and (c) cloud droplet number concentration $N_d$ for the polluted cases with $N_{\text{CCN}} = 1000\,\text{cm}^{-3}$. Respective mean profile (black solid line) and its standard deviation (error bar) are superimposed. For the polluted cases with $N_{\text{CCN}} = 100\,\text{cm}^{-3}$, only the mean profiles are shown (red solid lines). In both cases, the dashed profile is the theoretical adiabatic limit calculated for conditions at cloud base ($T_{\text{cb}} = 293\,\text{K}, 4\,\text{K}\,\text{km}^{-1}$) and $N_{\text{cb}} = 300\,\text{cm}^{-3}$ for the polluted and $N_{\text{cb}} = 50\,\text{cm}^{-3}$ for the clean case.*

– "Page 8, line 8: remove "the""

   → *Thanks, done.*

– "Page 8, line 11: add comma before "which""

   → *Done.*

– "Figure 4: Again, this figures lacks "a)" and "b)", and as far as I can tell, the "b)" part is not discussed at all in the text."

   → *Thanks for this note. We now have labeled each sub panel with "a)", "b)" and "c)". We now also discuss all sub plots in the revised manuscript:*

   *Page 10, Line 14ff now read:*

> *In Figure 4a, a three-dimensional visualization of the observation kernel method is presented. While the observation position (yellow dot) is moved through the model domain, the result of the convolution between observation kernel and cloud field is shown as arbitrary score on the surface in Figure 4a. A more detailed view on the observation kernel is given in Figure 4b by a horizontal and in Figure 4c by a vertical cut at the dashed cutting line. The arbitrary score is strongly negative in the vicinity of the observer to penalize locations where clouds are too close. Observation distances of 3 km to 5 km turned out to maximize the likelihood to observe a complete cloud side in the used LES model output. For a distance of 2 km and onward, the weighting score becomes thus positive with a maximum at 3.5 km to favor locations with clouds in this region. For all LES cloud fields on average, this method positions the observer at a distance of around 4 km from cloud sides.*

– "Page 10, line 4: Change "found" to "retrieved""

  → *Done.*

– "Page 10, Eq 5: change comma to full stop after the equation."

  → *Corrected.*

– "Page 11, line 4-9: As far as I can tell you don't show any cloud edge reff results. Also, no vertical gradients are shown to evaluate the "better agreement"."

→ *This is true. Since our technique to determine the apparent effective radius $\langle r_{\text{eff}}\rangle_{app}$ is essential for the realistic LES model output used (Figure 5b), we now also show results for the simple line-of-sight method in Figure 5a.*

*Page 13, line 16ff now read:*

> *For the cloud scene shown in Figure 5, Figure 6b shows the apparent effective radius $\langle r_{\text{eff}}\rangle_{app}$ obtained with MYSTIC REFF. Compared with the effective radius found at the cloud edge shown in Figure 6a, the apparent effective radius $\langle r_{\text{eff}}\rangle_{app}$ appears much smoother in Figure 6b. The range of values for $\langle r_{\text{eff}}\rangle_{app}$ obviously compares much better with the range of values of $r_{\text{eff}}$ shown in Figure 2(B3) and Figure 3a.*

[Figure]

**Figure 6. (a)** *Effective radii $r_{\text{eff}}$ found at cloud edge for the scene shown in Figure 5,* **(b)** *Apparent effective radii $\langle r_{\text{eff}}\rangle_{app}$ obtained with MYSTIC REFF for the same scene.*

– "Page 11, lines 14-15. This sentence sounds awkward and I am not sure what you mean here."

> → *Yes, the original sentence was too unspecific and confusing. For this reason we inserted an additional sentence to clarify what we meant with "In such a situation, ...":*
>
> *Page 14, line 10ff now read:*
>
> > *(...) In contrast to the typical observation geometry from above, where a plane-parallel cloud is assumed, the cloud surface orientation is mostly unknown for the cloud side perspective. In such a situation, where only the scattering angle $\vartheta_s$ is known, the limitation to optically thicker clouds can be a way out. (...)*

– "Page 11, line 16: add comma before "where""

> → *Corrected. This sentence now precedes the sentence mentioned in the last comment as clarification.*

– "Page 11, line 29: This seems to be a really high optical thickness. Why did you choose that? From a TOA retrieval perspective, this is not really realistic. For comparison: the operational MODIS retrieval stops at tau=150, which is rarely encountered. Does the change in perspective (i.e., cloud side) yield this large limit? Please clarify."

> → *Thank you for this comment. We already discussed this in our answer to one of your general comments ("optically thicker" vs. "optically thick"). As we mentioned there, we now dedicated a short subsection (Section 4.1, " Limitation to optically thicker clouds") to reason and specify our limitation to higher optical thickness.*

> *In Section 4.2 ("Ambiguities of reflected radiances"), we use a spherical water cloud with this high optical thickness of $\tau_c = 500$, to dissect the impact of an unknown cloud surface orientation on the effective radius retrieval. For the subsequent development of the gradient classifier (Section 4.3, "Additional information from surrounding pixels"), the shadow filter (Section 4.4, "Exclusion of cloud shadows") and for the retrieval itself (Section 5, "Retrieval") we use the RICO LES output with the described optical thickness range of $\tau_c = 15 - 150$. This retrieval specifications regarding optical thickness are now also mentioned and summarized in the conclusion (Section 7).*

– "Page 12, line 9: "optically thick clouds" is very ambiguous. A cirrus is optically thick if tau>5, a stratus with tau> 25, a cumulus with tau>50. Can you be more specific?"

> → *Thank you for this suggestion. We hope to have answered this question with our last reply and our answer to one of your general remarks ("optically thicker" vs. "optically thick"). In addition, we are now more specific and repeat our definitions throughout the manuscript when appropriate ("optically thicker", $\tau_c > 15$) and ("optically thick", $\tau_c = 500$).*

– "Figure 8: What is shown in the circles? What is the difference between grey and black lines? This Figure is very dense and includes a lot of information, it needs a better description.'

> → *Thank you for this question. Since referee #1 (AMTD-2018-234-RC1) also was confused about Fig. 8, we completely restructured this section and disentangled the plots from Fig. 8 (in the old manuscript version) into Fig. 11 and Fig. 12 (in the new manuscript version).*

*The gray circles/spheres in Figure 11a,c are radiance images $L_{0.87}$ of the spherical water cloud, while the colored spheres in Figure 11b,d are images of the radiance ratio $L_{2.10}/L_{0.87}$. We have chosen to show this ratio in Fig. 11 and the 2-wavelength diagram in Fig. 12 to connect the scattered points in Fig. 12 with their spatial location on the sphere in Fig. 11.*

*The large green and red dots in Fig. 11 and Fig. 12 indicate cloud surfaces with same local illumination angle $\vartheta_0^* = 30°$, but steeper ($\vartheta^* < \vartheta_0^*$, green dot) or more oblique local viewing angle ($\vartheta^* > \vartheta_0^*$, red dot). These two perspectives are illustrated in Figure 9. We now removed the gray lines (which too indicated results with same local illumination angle $\vartheta_0^*$) to reduce the complexity of this plot. Analogous to Fig. 8, the black isolines show radiances from one-dimensional DISORT simulations for optically thick water clouds with different effective radii.*

*We updated the captions of these figures to ease their understanding. Furthermore, the answer to #1 (AMTD-2018-234-RC1) is copied here to give a total overview of our changes:*

*In addition, we included more color cues and markings to separate the two considered cases ($\vartheta^* = \vartheta_0^*$, direct backscatter, orange color and $\vartheta^* \neq \vartheta_0^*$, not direct backscatter, blue color) in Fig. 7, 9, 11 and 12.*

*Description of Fig. 11 now reads (Page 19, line 7ff):*

Discussion Paper | Discussion Paper | Discussion Paper | Discussion Paper |

*For the direct backscatter geometry ($\vartheta^* = \vartheta_0^*$) on the left and outside the direct backscatter geometry ($\vartheta^* \neq \vartheta_0^*$) on the right, Figure 11a,c show radiance images of $L_{0.87}$ and Figure 11b,d show radiance ratios $L_{2.10}/L_{0.87}$ for the spherical water cloud. The colored radiance ratios will later help to identify regions on the sphere within the 2-wavelength diagram. Furthermore, the two viewing geometries considered in Figure 9 are marked by the green and red dots.*

*For the case with $\vartheta^* \neq \vartheta_0^*$, we furthermore included more markings to separate the two cases ($\vartheta^* > \vartheta_0^*$, oblique viewing angle, red color and $\vartheta^* < \vartheta_0^*$, steep viewing angle, green color) in Fig. 9 and 12.*

*Description of Fig. 12 now reads (Page 19, line 11ff):*

*Figure 12 shows the results in 2-wavelength diagrams for the direct backscatter direction (left) and for a scattering angle of 150° (right). In the 2-wavelength diagrams the radiance pairs from the 3D MYSTIC simulation are shown as scattered points, the results from the one-dimensional DISORT simulations for different effective radii are shown as black lines. Just like in 9 to 11, the large green and red dots in Figure 12 indicate cloud surfaces with same local illumination angle $\vartheta_0^* = 30°$, but steeper ($\vartheta^* < \vartheta_0^*$, green dot) or more oblique local viewing angle ($\vartheta^* > \vartheta_0^*$, red dot).*

*The same (red/green) color scheme is now also used in the following section (4.2.1 Comparison of pixel brightness) to strengthen the connection between the preceding analysis of radiance ambiguities and the mitigation of radiance ambiguities in Section 4.2.1.*

**–** "Page 13, last sentence: remove "Figure""

> → *Removed.*

**–** "Page 15, lines 9-10: This sentence is awkward."

> → *Thanks for this comment. This sentence now reads:*
>
> *Page 21, line 20-21:*
>
>> *Previous studies, like Marshak et al. (2006a) and Zinner et al. (2008), did not investigate in detail the origin of these radiance ambiguities.*

**–** "Page 15, line 12: change the last "," to "and" or "to"."

> → *Thanks, changed.*

**–** "Page 15, line 14: add comma after "Here""

> → *We added this comma after "Here"*

**–** "Page 15, line 16: pixels (plural)"

> → *Fixed.*

**–** "Page 16, line 11: define CMOS"

> → *We changed "CMOS" to "image sensor" since the suppression of pixel-to-pixel noise has to be considered for all image sensors.*

[revised manuscript text omitted]

$$2\underline{r_{\text{eff}}^{\text{flip}}} \quad = 16\,\mu\text{m} \ -r_{\text{eff}}^{\text{orig}} \qquad (\text{CCN} = 1000\,\text{cm}^{-1})$$

$$r_{\text{eff}}^{\text{flip}} \quad = 26\,\mu\text{m} \ -r_{\text{eff}}^{\text{orig}} \qquad (\text{CCN} = 100\,\text{cm}^{-1})$$

10  fields and add an offset $r_{\text{eff}}^{\text{offset}}$:

$$r_{\text{eff}}^{\text{flip}} = -r_{\text{eff}}^{\text{orig}} + r_{\text{eff}}^{\text{offset}}. \tag{17}$$

 To ensure positive and realistic values for $r_{\text{eff}}^{\text{flip}}$, the offset $r_{\text{eff}}^{\text{offset}}$ was chosen to be at least 4 µm larger than the  largest values found in  all cloud fields. Thus, $r_{\text{eff}}^{\text{offset}} = 12\,\mu\text{m} + 4\,\mu\text{m} = 16\,\mu\text{m}$ was used in Equation (17) for the polluted cloud ensemble (CCN $=1000\,\text{cm}^{-3}$)

15  and $r_{\text{eff}}^{\text{offset}} = 22\,\mu\text{m} + 4\,\mu\text{m} = 26\,\mu\text{m}$ for the clean cloud ensemble (CCN $=100\,\text{cm}^{-3}$). To preserve the optical thickness $\tau^{\text{orig}}$ of the original cloud field ,

$$\tau^{\text{flip}} \equiv \tau^{\text{orig}}, \tag{18}$$

the well established relationship

$$\underline{\tau} \qquad \propto \frac{\text{LWC}}{r_{\text{eff}}}$$

20  $$\tau^{\text{flip}} \qquad \equiv \tau^{\text{orig}}$$

$$\text{LWC}^{\text{flip}} \quad = \frac{r_{\text{eff}}^{\text{flip}}}{r_{\text{eff}}^{\text{orig}}}\text{LWC}^{\text{orig}}$$

in Equation (19) was used to derive the liquid water content LWC$^{\text{flip}}$ for the flipped cases in Equation (20):

$$\tau \propto \frac{\text{LWC}}{r_{\text{eff}}}, \tag{19}$$

[revised manuscript text omitted]